# Adversarial Learning for Feature Shift Detection and Correction

**Míriam Barrabés** [1,2,*]   **Daniel Mas Montserrat** [1,*]   **Margarita Geleta** [3]
**Xavier Giró-i-Nieto** [4,†]   **Alexander G. Ioannidis** [1‡]
[1]Stanford University   [2]Universitat Politècnica de Catalunya
[3]University of California, Berkeley     [4]Amazon

## Abstract

Data shift is a phenomenon present in many real-world applications, and while there are multiple methods attempting to detect shifts, the task of localizing and correcting the features originating such shifts has not been studied in depth. Feature shifts can occur in many datasets, including in multi-sensor data, where some sensors are malfunctioning, or in tabular and structured data, including biomedical, financial, and survey data, where faulty standardization and data processing pipelines can lead to erroneous features. In this work, we explore using the principles of adversarial learning, where the information from several discriminators trained to distinguish between two distributions is used to both detect the corrupted features and fix them in order to remove the distribution shift between datasets. We show that mainstream supervised classifiers, such as random forest or gradient boosting trees, combined with simple iterative heuristics, can localize and correct feature shifts, outperforming current statistical and neural network-based techniques. The code is available at `https://github.com/AI-sandbox/DataFix`.

## 1   Introduction

Distribution shifts in multi-dimensional data, caused by one or more "corrupted" dimensions, are common in various real-world applications. Data streams from multi-sensor environments in fields like medicine, industry, finance, and defense, can experience shifts due to faulty sensors [1]. Tabular and structured data used in domains such as economics, biology, genomics, and social sciences, can encounter distribution shifts caused by improper standardization, erroneous data processing, data collection procedures, or human entry errors [2, 3]. Combining databases from diverse sources is common in many fields: various hospitals collect different phenotypic characteristics from patients, and governmental institutions often capture different socio-economic indicators of citizens. However, proper standardization of data is seldom applied across data sources, and merging these poses challenges that often require data-dependent and domain-specific techniques [4]. For example, efficient computational techniques are used to process high-dimensional genomic sequences [5], and domain knowledge is incorporated in processing data from social sciences [6]. This process typically involves changing, imputing, merging, and removing both features and samples, potentially leading to complex pipelines that can originate distribution shifts between datasets. These shifts can create erroneous scientific results if present in datasets used for medical or social studies, or indicate infrastructure failure if present in sensory data in industrial applications. Therefore, detecting, localizing, and correcting such shifts is an important task that remains a relatively unsolved problem.

The "localization" of feature shifts is the task of finding which features (i.e., dimensions) of the dataset are causing the shift within multiple sets of data. This step can be critical in order to address the error

---

[*]Equal contribution.

[†]Work done prior to joining Amazon.

[‡]Correspondence to A.G.I. [ioannidis@stanford.edu].

37th Conference on Neural Information Processing Systems (NeurIPS 2023).

source, by physical intervention in the multi-sensory scenario, or by data removal or fixing in tabular data-based applications. There has been extensive work on distribution shift detection and anomaly detection [7, 8], however, such methods focus mainly on detecting if two sets of data follow the same distribution, or on detecting outliers, and the task of identifying the exact components generating the shift remains partially unexplored. Recently, work combining machine learning techniques with statistical testing provided state-of-the-art results showing that localizing corrupted features can be done accurately [9]. In this work, we explore "feature selection" methods as a mechanism to detect potential feature shifts. The "correction" of feature shifts consists of replacing corrupted feature values with new ones, ensuring that the updated dataset follows a distribution that is equal to, or more similar to, the distribution without corruptions. This can be useful in data homogenization and quality control pipelines, enabling the elimination of shifts between multiple data sources and their combination for downstream applications, including data discovery or machine learning training. Feature shift correction can be framed as a supervised problem, employing regression or classification methods to predict the new feature values; as a missing data imputation problem, by treating corrupted features as missing values [10, 11]; or as a distribution alignment problem [12, 13], where mappings between distributions are learned using adversarial learning, optimal transport, or statistical divergences. Effective techniques for correcting distribution shifts, either by learning parametric models to predict new values for corrupted features, or directly updating values without explicit model learning, can be very useful to avoid incorrect or biased results in analysis based on data containing feature shifts. We provide a more formal definition of the feature shift detection and correction tasks in the following sections.

Tools for automated shift detection, data monitoring, and feature analysis and correction are becoming more common in data quality control, data homogenization, data processing pipelines, data-centric AI, MLOps, and deployment monitoring for ML Systems [14, 15, 16, 17]. In this work, we introduce "DataFix", a framework that makes use of discriminators trained to distinguish samples coming from two different distributions, in order to estimate distribution shifts, localize which features are causing them, and modify the samples in order to reduce or remove the shift. DataFix is composed of two systems: "DF-Locate", which locates the features causing the distribution shift, and "DF-Correct", which directly modifies the samples in order to decrease the distribution shift between two datasets. DF-Locate consists of an iterative process where a discriminator combined with a feature importance estimator is used to identify the most discriminative features between samples coming from two different distributions. The detected features are removed from the dataset, a new discriminator is trained, new feature importances are estimated, and the process is repeated until no distribution shift is detected. DF-Correct replaces the values of the corrupted features detected by DF-Locate with values that exhibit a minimal probability of being corrupted when processed by a discriminator.

Our contributions are as follows: (1) we motivate, define, and formalize the problem of feature shift detection and correction and relate it to feature selection and adversarial learning frameworks; (2) we propose an iterative algorithm that makes use of feature selection techniques from discriminators to accurately detect manipulated features; (3) we propose an iterative algorithm that makes use of a discriminator to guide the correction of a corrupted dataset in order to decrease its distribution shift; (4) we provide an in-depth experimental evaluation with multiple manipulation types and datasets.

## 2 Related Work

**Distribution Shift Detection and Localization.** Distribution shift detection involves identifying if $p \neq q$, where $p$ and $q$ are the reference and query distributions respectively. While there is extensive research on detecting distribution shifts in univariate distributions [18, 19, 7], exploration in the multivariate setting is relatively limited [20]. Multivariate distributions can exhibit various types of shifts, including covariate, label, concept, or marginal shifts, among others [21, 22, 23]. The work in [8] applies hypothesis testing for concept drift detection. In [20], two-sample hypothesis testing is explored for shift detection, and a comparison is made between multivariate hypothesis tests via Maximum-Mean Discrepancy (MMD) [24], univariate hypothesis tests with marginal Kolmogorov-Smirnov (KS) tests and the Bonferroni correction [25], and dimensionality reduction techniques, among others. However, these works focus on detecting if a distribution shift is present but do not localize which features are causing it. Recently, a conditional test method was able to identify the features originating the shift [9] with model-free and model-based approaches: K-Nearest Neighbors with KS statistic (KNN-KS), multivariate Gaussian with KS (MB-KS), multivariate Gaussian and

Fisher-divergence test statistics (MB-SM), and deep density neural network models with Fisher-divergence test (Deep-SM). However, performing feature shift localization remains challenging.

**Feature Selection.** Feature selection methods can remove redundant features affecting efficiency or performance, and find the most relevant features, providing interpretability. Common techniques include: filtering [26, 27], wrapper [28, 29] and embedded methods [30, 31]. Filtering methods, including univariate Mutual Information statistics (MI) [32], ANOVA-F test [33], and Chi-square test [34], rank features based on data statistics. Minimum Redundancy Maximum Relevance (MRMR) [35, 36] selects relevant features while minimizing redundancy with the selected ones. Fast-Conditional Mutual Information Maximization (FAST-CMIM) [37] selects features that maximize MI, conditional to previously selected features. Wrapper methods select feature subsets by training models on them and adding and removing features through a search process that can be computationally intensive. Embedded methods use the inherent scores computed by predictive models, such as logistic regression weights [38] or the mean decrease of impurity (Gini index) in random forests [39]. Most methods select the features with the highest importance scores or all that exceed a given threshold [40].

**Missing Data Imputation.** Missing data imputation methods can remove distribution shifts by considering the shifted features as missing and re-constructing them. Imputation methods range from simple record deletion, zero imputation, mean imputation, and deck imputation [41], to machine learning-based methods that apply regression and classification to reconstruct missing data including MICE [10], MissForest [11], and Matrix Completion [42, 43]. Some generative approaches include Expectation Maximization algorithms [44] and deep learning-based methods such as the MIWAE autoencoder [45]. Causality-based techniques such as MIRACLE have proven to be highly accurate [46]. HyperImpute [47] provides state-of-the-art imputation by using an iterative process with ML methods, including gradient boosting machines and neural networks trained to impute missing data.

**Optimal Transport, Distribution Alignment, and Adversarial Learning.** Feature shift correction can also be performed with distribution alignment methods, optimal transport, and adversarial learning. Distribution alignment and representation learning methods learn a mapping between distributions and can be used to remove shifts by projecting the shifted samples into the non-shifted distribution. Iterative Alignment Flows [12] and Deep Density Destructors [13] are neural network examples that map samples between distributions. Optimal transport-based methods reduce the Wasserstein distance between distributions. The work in [48] applies a Sinkhorn-based optimization process to perform imputation. Adversarial learning has been explored in generative adversarial networks such as GAIN [49], where the loss of a discriminator is used to improve imputation accuracies.

**Data-centric AI.** Data-centric AI focuses on systematically enriching data quality and quantity as a means to boost AI and ML performance. It covers strategies that impact every phase of the data lifecycle, from data collection [50], labeling [51, 52], augmentation [53, 54, 55], and integration [56], to the crucial processes of data cleaning [57, 58], feature extraction [59], and transformation [60]. These include programmatic automation strategies, which rely on programs guided by heuristics and statistics for automatic data processing [61, 62, 63], as well as learning-based automation techniques that optimize data automation procedures, typically using machine learning [64, 65, 66]. The detection and removal of distribution shifts are becoming essential steps of the data-centric AI toolbox [17].

## 3 Proposed Framework

**Definition 1. [Feature Shift]** *We are given two sets of $d$-dimensional samples $X = \{x_1, x_2, ..., x_N\}$ and $Y = \{y_1, y_2, ..., y_N\}$, with $x_i, y_i \in \mathbb{R}^d$, from distributions $p$ and $q$, respectively. A feature shift between $p$ and $q$ occurs when $D(p, q) > \varepsilon$ and $D(p_S, q_S) \leq \varepsilon$, where $D$ is a valid divergence or distance between distributions, $S$ and $C$ are the subsets of non-corrupted and corrupted features respectively, such that $|S \cup C| = d$, and $p_S$ and $q_S$ are the distributions restricted to $S$.*

We will refer to $X$ and $Y$ as the "reference" and "query" datasets, and to $p$ and $q$ as the "reference" and "query" distributions, respectively. We will assume that the reference contains only "non-corrupted" features, while the query contains one or more "corrupted" dimensions that we will want to detect and correct. Here we consider scenarios with $\varepsilon = 0$, $D(p_S, q_S) = 0$, $D(p, q) \gg 0$, and $|S| > |C| \geq 1$. That is, there are more non-corrupted than corrupted features, the divergence becomes 0 if the corrupted features are removed, and is large enough to be empirically detected otherwise. We will consider multiple types of distribution shifts: marginal shifts with $D(p_i, q_i) > \varepsilon$, where $p_i$ and $q_i$ represent the marginal distribution of the $i$th dimension resulting from additive and non-linear

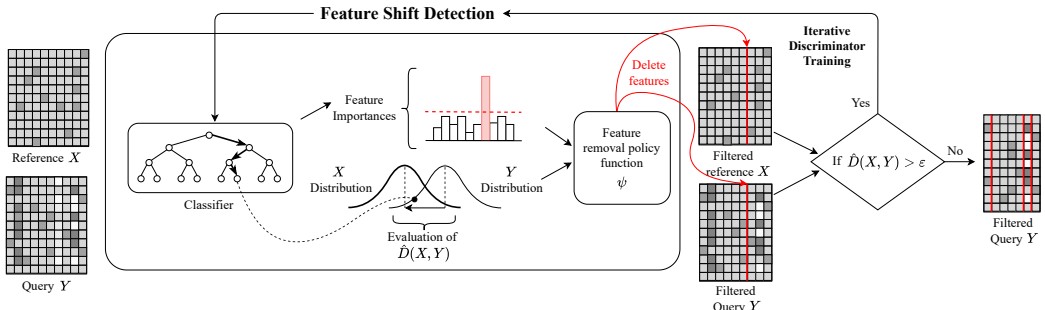

Figure 1: DF-Locate overview diagram.

transformations; correlation shifts where $D(p, q) > \varepsilon$ but $D(p_i, q_i) \leq \varepsilon$ for all $i$; and correlation shifts where $D(p_S, q_S) \leq \varepsilon$ and $D(p_C, q_C) \leq \varepsilon$ but $D(p, q) > \varepsilon$. In the latter case, correlations are maintained locally, but a shift is present when considering all features simultaneously.

**Definition 2. [Feature Shift Localization Task]** *The task of localizing a feature shift consists of finding the smallest subset of features $C$, with $\overline{C} = S$, that satisfies $D(p_{\overline{C}}, q_{\overline{C}}) \leq \varepsilon$, that is $C = \operatorname{argmin}_{D(p_{\overline{C}}, q_{\overline{C}}) \leq \varepsilon} |C|$.*

The number of corrupted features $|C|$ will be assumed to be unknown a priori. Furthermore, because the distributions $p$ and $q$ are assumed to be unknown and only their samples are accessible, the task needs to be approximated, requiring careful consideration of trade-offs such as falsely flagging non-corrupted features (false positives) and failing to detect corrupted ones (false negatives).

**Definition 3. [Feature Shift Correction Task]** *The task of correcting a feature shift consists of finding a new matrix $Y' = \{y_1', y_2', ..., y_N'\}$ such that $y_i' \sim q'$ and $D(p, q') \leq \varepsilon$, while keeping the non-corrupted features unchanged, that is $Y_S' = Y_S$.*

The correction of shifted features can be done through parametric models that perform sample-wise transformations $y' = \phi(y)$, prediction or imputation methods $y_C' = \phi(y_{\overline{C}})$, or optimization procedures or heuristics $Y' = \Phi(X, Y)$. Our proposed approach falls in the latter category.

## 4 Feature Shift Detection: DF-Locate

DF-Locate (Fig. 1, Algorithm 1) employs an iterative process to detect the presence of a shift and determine the features causing it. At each iteration, a classifier is trained to detect the origin (binary label indicating reference vs query) of each sample, the output predictions are used to estimate the divergence between distributions, and the feature importance scores provided by the binary classifier are used to locate the features originating the shift. At the end of each iteration, the features detected as corrupted are removed, and the process is repeated until no divergence is detected.

**Shift Detection.** We detect whether a shift is present by estimating an $f$-divergence between the distributions $p$ and $q$ by using an empirical approximation of the variational form [67, 68, 69]:

$$\hat{D}_\theta(X, Y) = \frac{1}{N_x} \sum_{i=1}^{N_x} f'(r_\theta(x_i)) - \frac{1}{N_y} \sum_{j=1}^{N_y} f^*(f'(r_\theta(y_j))) \tag{1}$$

where $X = \{x_1, ..., x_N\}$ and $Y = \{y_1, ..., y_N\}$, with $x_i \sim p$, $y_j \sim q$, and $r_\theta(x)$ is a function approximating the likelihood ratio between $p$ and $q$, obtained by training a binary classifier $\mathcal{D}_\theta(x)$ such that $r_\theta(x) = \mathcal{D}_\theta(x)/1 - \mathcal{D}_\theta(x)$. We use 5-fold train-evaluation such that 80% of the samples are used to train a random forest binary classifier $\mathcal{D}_\theta(x)$, and the 20% left is used to estimate the empirical divergence with $N_x$ and $N_y$ testing samples of the reference and query datasets, respectively. The resulting estimates are averaged across folds to reduce variance. The function $f$ is the generator function of the $f$-divergence, $f'$ is its first order derivative, and $f^*$ is its Fenchel conjugate [70]. By changing $f$, various divergences can be obtained such as the Kullback-Leibler divergence, the Jensen-Shannon divergence used in GANs, or the total variation distance, among others.

The true $f$-divergence is recovered in expectation $D_f(p,q) = \mathbb{E}_{x \sim p, y \sim q}[\hat{D}_\theta(X, Y)]$ when $r_\theta(x) = r^*(x) = p(x)/q(x)$ is the true likelihood ratio function, defining the Bayes decision rule that optimally separates $p$ and $q$. In this work, we make use of the total variation distance $0 \leq D_{TV}(p,q) \leq 1$, defined by setting $f'(u) = (f^* \circ f')(u) = \frac{1}{2}\text{sign}(u-1)$. The discrete nature of $f'(u)$ makes the empirical estimator robust to poorly calibrated classifiers. Furthermore, its value is proportional to the empirical balanced accuracy on the evaluation test. Intuitively, if $\hat{D}_{TV}(X, Y) > 0$, or equivalently, the test balanced accuracy is larger than $0.5$ (random chance), it might indicate that $p \neq q$. After detecting the presence of a shift, the next step is to localize the features originating it.

**Feature Selection for Shift Localization.** Our proposed method employs feature importance and feature selection techniques in order to detect the features originating a distribution shift. These techniques often involve approximating the following estimation problem, either implicitly or explicitly:

$$C = \underset{|C| \leq k}{\text{argmax}} \, I(z_C; t) \tag{2}$$

where $C$ is the feature subset of size up to $k$, which maximizes the mutual information $I$ between the input sample $z$, restricted to the features in $C$, and the label $t$. By defining $z$ as the samples coming from $X$ and $Y$, and $t$ as the binary label of origin, such that $z|_{t=0} \sim p$ and $z|_{t=1} \sim q$, Fano's inequality [71] and LeCam's method [72] can be used to relate the mutual information between $z$ and $t$ and the total variation distance between $p$ and $q$:

$$1 - H_2\left(\frac{D_{TV}(p,q) + 1}{2}\right) \leq I(z;t) \leq D_{TV}(p,q) \tag{3}$$

where $H_2$ is the entropy with base-2 logarithm. If the number of manipulated features is known $|C| = k$, and $D_{TV}(p,q) = D_{TV}(p_C, q_C) = 1$, then $C = \text{argmax}_{|C| \leq k} I(z_C; t) = \text{argmin}_{D_{TV}(p_{\overline{C}}, q_{\overline{C}})=0} |C|$, thereby rendering the problem of feature selection and feature shift detection equivalent. See Section C for further discussion. In practice, the distributions are unknown making Eq. 2 intractable and most methods predict a score that approximates the amount of information relative to the label present at each feature: $\beta = F(Z, T)$, where $\beta \in \mathbb{R}^d$ and $F(\cdot)$ is a function mapping samples $Z$ and labels $T$ to a feature-wise importance score. Here, we use the mean decrease of impurity from the random forest classifier $\mathcal{D}_\theta$ as the feature importance scores.

**Feature removal policy.** We introduce the feature removal policy function $\psi$ as an heuristic to approximate Eq. 2, which utilizes the predicted feature importances $\beta$ and the estimated total variation distance $\hat{D} = \hat{D}_{TV}(X, Y)$ to select likely corrupted features $C = \psi(\beta, \hat{D})$. First, $\beta$ is normalized $\beta' = \frac{|\beta|}{\sum_{i=0}^{d-1} |\beta_i|}$ and sorted $\beta'_\pi$, such that $\beta'_{\pi(0)} \geq \beta'_{\pi(1)} \geq ... \geq \beta'_{\pi(d-1)}$. Given the cumulative sum $\gamma(k) = \sum_{j=0}^{k} \beta'_{\pi(j)}$, the sorted features from 0 to $J$ are selected as corrupted, where $J$ is the smallest index such that $\gamma(k) \geq \tau \hat{D}$ (Eq. 4), and $\tau$ is a hyperparameter set by hyperparameter optimization. Finally, $\psi$ returns the features from 0 to $J$ with scores higher than $\frac{1}{d}$ (Eq. 5):

$$J(\tau, \hat{D}, \beta) = \underset{k; \gamma(k) \geq \tau \hat{D}}{\text{argmin}} \, \gamma(k) \quad (4) \qquad C = \{\pi^{-1}(j) : 0 \leq j \leq J, \beta'_{\pi(j)} > \frac{1}{d}\} \quad (5)$$

Note that $\tau \hat{D}$ acts as a threshold to select how many features are selected as corrupted. If the threshold is small, $\psi$ simply returns the feature with the highest feature importance, while a larger threshold will make $\psi$ return more features. By defining the threshold as the product of $\tau$ and $\hat{D}$, we ensure that when the shift is small (low $\hat{D}$), a smaller number of features are flagged as corrupted.

**Iterative Process.** DF-Locate performs an iterative process that starts with the reference $X^{(0)} = X$ and query $Y^{(0)} = Y$ datasets. At each iteration $i$, it trains a set of discriminators $\theta^{(i)} = \text{argmax}_\theta \hat{D}_{TV}(X^{(i)}, Y^{(i)})$ consisting of random forest binary classifiers. The classifier predictions are used to estimate $\hat{D}^{(i)} = \hat{D}_{TV}(X^{(i)}, Y^{(i)})$ and, combined with the Gini importance from the random forests $\beta^{(i)} = F(X^{(i)}, Y^{(i)})$, a set of corrupted features are localized using the feature removal policy $C_i = \psi(\beta^{(i)}, \hat{D}^{(i)})$. The selected features $C_i$ are removed from the dataset, such that $X^{(i+1)} = X^{(i)}_{\overline{C_i}}$ and $Y^{(i+1)} = Y^{(i)}_{\overline{C_i}}$, and the process is repeated while $\hat{D}_{TV}(X^{(i)}, Y^{(i)}) > 0.02$ or

until half of the features have been removed or no features are removed at the current iteration. After $l$ iterations, the set of corrupted features is obtained as $C' = \bigcup_{i=1}^{l} C_i$. Finally, a refinement stage (see below and Section E.3) predicts the final set $C$. More details are provided in Section E.

**Refinement stage.** DF-Locate stores all intermediate steps, including the indexes of the predicted corrupted feature locations and the estimated $\hat{D}$ at each iteration, in order to revisit the iterative filtering process and select the optimal stopping point. To determine the optimal iteration, the elbow or knee is found from a processed curve depicting the empirical total variation distance as a function of the total number of removed features (see Section E.3). This refinement stage effectively eliminates false positives and enhances the accuracy of feature shift localization.

---

**Algorithm 1** DF-Locate

1: **Inputs:**
    $X$;                                     ▷ Reference
    $Y$;                                     ▷ Query
    $\tau$;                           ▷ Feature Selection Threshold
    $\epsilon$;                              ▷ Divergence Threshold
2: $X^{(0)} = X$
3: $Y^{(0)} = Y$
4: $i = 0$
5: $k^{(0)} = 0$
6: **while** $\hat{D}_\theta(X^{(i)}, Y^{(i)}) > \epsilon$ and $k^{(i)} < \frac{|Y|}{2}$ and $k^{(i)} - k^{(i-1)} > 0$ **do**
7:     $\theta \leftarrow \text{Train}(X^{(i)}, Y^{(i)})$      ▷ Train discriminator
8:     $\hat{D} \leftarrow \hat{D}_\theta(X^{(i)}, Y^{(i)})$      ▷ Estimate divergence
9:     $\beta \leftarrow F_\theta(X^{(i)}, Y^{(i)})$      ▷ Estimate feature importance
10:     $C_i \leftarrow \psi_\tau(\beta, \hat{D})$         ▷ Select corrupted features
11:     $X^{(i+1)}, Y^{(i+1)} \leftarrow X^{(i)}_{\overline{C_i}}, Y^{(i)}_{\overline{C_i}}$      ▷ Remove detected features
12:     $k^{(i+1)} \leftarrow k^{(i)} + |C_i|$ ▷ Update detected feature counter
13:     $i \leftarrow i + 1$
14: **end while**
15: $C' = \bigcup_{j=0}^{i-1} C_j$            ▷ Combine all detected features
16: $C \leftarrow \text{Refine}(C')$            ▷ Refine detected features
17: **return** $C$

---

**Algorithm 2** DF-Correct

1: **Inputs:**
    $X$;                                     ▷ Reference
    $Y$;                                     ▷ Query
    $C$;                             ▷ Corrupted Features
    $\epsilon$;                              ▷ Divergence Threshold
2: $V = \{Y^0, Y^1, Y^2\} \leftarrow \text{Impute}(X, Y, C)$
3: $Y' \leftarrow \text{argmin}_{Y^i \in V} \hat{D}_\theta(X, Y^i)$
4: **if** $\hat{D}_\theta(X, Y') < \epsilon$ **then**
5:     **return** $Y'$
6: **end if**
7: **for** epoch **do**
8:     $\theta \leftarrow \text{Train}(X, Y')$            ▷ Train discriminators
9:     $L \leftarrow \text{DetectIncorrect}(\mathcal{D}_\theta(Y'))$      ▷ Detect samples that require feature correction
10:     $B \leftarrow \text{GenerateProposals}(X, Y', V, C)$
11:     **for** $i \in L$ **do**
12:         $b_i \leftarrow \text{argmax}_{b \in B}\, r_{\theta_i}(y_i^{(b)})$   ▷ Find best proposal
13:         $Y' \leftarrow \text{update}(Y', y_i^{(b_i)})$       ▷ Dataset with updated sample
14:     **end for**
15:     **if** $\hat{D}_\theta(X, Y') < \epsilon$ **then**
16:         **return** $Y'$
17:     **end if**
18: **end for**
19: **return** $Y'$

---

## 5 Feature Shift Correction: DF-Correct

After the set $C$ of features originating the shift has been detected by DF-Locate, DF-Correct (Figure 2, Algorithm 2) is applied to generate a new query dataset $Y'$ that rectifies the distribution shift. Ideally, the objective is to find $Y' \sim q'$ such that:

$$Y' = \underset{Y \sim q'; ||Y_{\overline{C}} - Y'_{\overline{C}}|| = 0}{\text{argmin}} D(p, q') \qquad (6)$$

where $D$ is a valid statistical divergence. Because $p$ and $q'$ are unknown, we approximate the optimization problem by using a discriminator $\mathcal{D}_\theta$ to predict the empirical estimate of an $f$-divergence:

$$Y' = \underset{Y}{\text{argmin}} \max_{\theta} \hat{D}_\theta(X, Y) \qquad (7)$$

This approach parallels the adversarial minimax optimization setting adopted in GANs, where samples are generated by a generator $Y' = \mathcal{G}_\omega(u)$, and a discriminator tries to accurately classify the generated samples as "fake". This training leads to the minimization of the Jenson-Shannon divergence, and the setting for GAN training can be generalized to any $f$-divergence [68]. Here, instead of training a generator, we directly update the corrupted values of the query dataset by trying to minimize the predicted likelihood of the updated samples coming from $q$, approximated with the discriminator $\mathcal{D}_\theta$. While neural network-based discriminators allow for direct optimization of the values of $Y$ through backpropagation, these can have suboptimal performance when classifying tabular and similar structured data, which is the focus of our work. Hence, we employ tree-based

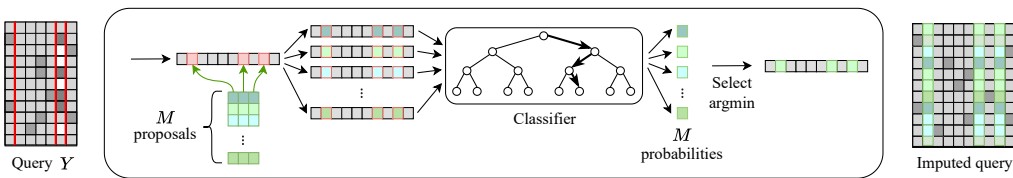

Figure 2: DF-Correct overview diagram.

techniques instead. Despite not being differentiable, derivative-free optimization and search heuristics can be used to update $Y$.

Two important aspects need to be taken into consideration: (a) the allowed search space for the values of $Y$, and (b) the frequency of training or updating the discriminator. In GANs, the space of possible values for generated samples is restricted by the complexity of the generator network, and, typically, the discriminator is updated in sync with each generator update. In our setting, if we allowed the search space to be the whole Euclidean space without updating or retraining the discriminator, we would be effectively conducting an adversarial attack [73, 74], where the empirical divergence would decrease while the true divergence would not. However, frequent retraining of the discriminator is computationally infeasible for tree-based techniques due to their lack of online training capabilities, necessitating retraining from scratch. Therefore, our discriminator is retrained only once at each iteration within our iterative process. Furthermore, we limit the search space for the possible values of the features in $Y'$ to a set of proposal values $B$, generated from the reference dataset $X$.

**Initial Imputation.** DF-Correct starts by setting the features C of the query dataset, previously detected as corrupted by DF-Locate, as missing. Initial missing data imputation is then performed with three distinct techniques: KNN, linear regression, and random sampling from the reference dataset $X_C$. This yields three imputed query datasets. A set of discriminators consisting of binary CatBoost [75] classifiers are trained for each reference and imputed query pair, and the empirical total variation distance is estimated following the same procedure as in DF-Locate. The imputed query dataset providing the lowest empirical divergence is selected as a starting point for the iterative process of DF-Correct. If the initial empirical divergence is already lower than $\varepsilon = 0.1$, the correction process is finalized. If not, DF-Correct applies the iterative process described below.

**Iterative Process.** The reference, the imputed query, and augmented samples (see Section F) are used to train $k = 2$ CatBoost binary classifiers to discriminate between reference and query datasets. $k$-fold splitting of the datasets is used to ensure that each classifier does not see the same sample during training and inference. The set of samples to be corrected $L$ is obtained by selecting $|Y|/2$ samples from $Y$ with the highest probability of being corrupted (or equivalently, the lowest probability of being from the reference distribution). Then, we generate a set of new feature value proposals $B = \{b_1, ..., b_N\}$, with $b_j \in \mathbb{R}^{|C|}$. This set comprises all the feature values within positions $C$ from the reference dataset $X$ and the imputed query with linear regression, alongside random permutations of the reference values. In other words, each $b \in B$ contains $|C|$ values obtained from $X$ (and imputed $Y$) that can be a potential replacement for the corrupted features of each query sample $y \in L$. Note that the size of $B$ is proportional to the size of $X$. Then, for every query sample $y \in L$ classified as "corrupted" by the discriminator, we replace the shifted features by all candidate values in $B$, and select the $b_i$ that provides the highest empirical likelihood of being a non-corrupted sample:

$$b_i = \underset{b \in B}{\arg\max}\, r_{\theta_i}(y_i^{(b)}) \tag{8}$$

where $y_i^{(b)}$ is the query sample $y_i$ with the corrupted features $C$ replaced with the values of $b$ such that $y_{iC} = b$, and $r_{\theta_i}$ is the likelihood ratio function from the classifier used to process the $i$th sample. In other words, we evaluate each sample $y \in L$ a total of $|B|$ times with the discriminator and select the $b$ providing the highest probability that $y^{(b)}$ is from the reference distribution. Then, we update the current corrected query dataset $Y'^{(k)}$ by replacing the corrupted sample $y_i$ with $y_i^{(b_i)}$, such that $Y'^{(k+1)} = (Y'^{(k)} \setminus y_i) \cup y_i^{(b)}$. By replacing corrupted features with values from the Eq. 8, the empirical divergence is decreased such that $\hat{D}_\theta(X, Y'^{(k+1)}) \leq \hat{D}_\theta(X, Y'^{(k)})$. After updating all the

"corrupted" features within the query dataset, the classifiers are retrained and the process is repeated until no divergence is detected or until the empirical divergence stops decreasing. The iterative process of replacing corrupted features, which reduces the empirical divergence, and retraining discriminators, which allows the estimation of more accurate likelihoods and divergences, approximates the minimax optimization process in Eq. 7. More details are in Section F.

## 6    Experimental Results

**Real world datasets.** We use multiple datasets including UCI datasets such as Gas [76], Energy [77], and Musk2 [78], OpenML datasets including Scene [79], MNIST [80], and Dilbert [81], datasets with DNA sequences such as Founders [82] and a private Dog DNA dataset (Canine), a subset of phenotypes from the UK Biobank (Phenotypes) [83], Covid-19 data [84], and simple datasets including values generated from Cosine and Polynomial functions. The datasets range from 8 to 198,473 features, and from 1,444 to 70,000 samples, including continuous and categorical datasets. We normalize each feature to have values from 0 to 1. We randomly divide each dataset into two equally sized subsets, corresponding to the reference $X$ and query $Y$ samples. Multiple manipulations are applied to randomly selected features of the query dataset. A detailed description of the datasets and the pre-processing applied is available in Section B.1.

**Feature Shift Manipulations.** We apply 10 different manipulations to the real world datasets in order to generate various distribution shifts. Table 1 describes each manipulation. Manipulations 1, 2, 4, 5, 6, and 7 distort the marginal distributions, with manipulation 4 leaving the mean approximately unchanged. Manipulations 4 and 6 have different levels of strength, controlled by parameters $\alpha$ and $\rho$, respectively. Manipulations 3 and 6 shuffle the feature values across samples, leaving the marginal distributions unchanged ($p_i = q_i$) but changing their correlations. Manipulation 3 performs a different random permutation at each feature, removing all correlation between features, while manipulation 6 performs the same permutation for all features, such that $p_C = q_C$, but $p \neq q$. Manipulations 9 and 10, applied to continuous and categorical variables respectively, replace the features with values predicted by k-nearest neighbor (KNN) trying to reconstruct the corrupted features. The nature of the shifts originated by KNN will depend on the given dataset and distribution. We discuss the nature of shifts originated by predictive and imputation models in Section D. Manipulations are applied to continuous features, categorical features, or both. In total, we apply 10 manipulations on continuous features and 8 manipulations on categorical features. Each manipulation is applied to 5%, 10%, and 25% of the features in the query. This produces a reference dataset with 24 and 30 query datasets for categorical and continuous data, respectively. Each query dataset corresponds to a distinct transformation applied to a specific fraction of the features.

Table 1: Manipulation types applied to continuous and/or categorical features.

| Type | Mapping | Description | Shift | Data |
|------|---------|-------------|-------|------|
| 1 | $x \sim \text{Uniform}(0,1)$ | Each value is substituted by a random number between 0 and 1. | $p_i \neq q_i$ | Cont. |
| 2 | $1 - x$ | Each value is negated. | $p_i \neq q_i, \mathbb{E}[q_i] = 1 - \mathbb{E}[p_i]$ | Both |
| 3 | $P_i X_i$ | $P_i$ is a random permutation matrix applied to feature $i$. | $p_i = q_i, p_C \neq q_C,$ $q_C = \prod_{i \in C} q_i$ | Both |
| 4.1-4.3 | $\text{clamp}_{0,1}(x + \alpha \sigma)$ $\sigma \sim \text{Rademacher}(0.5)$ | Add constant noise with a random sign. $\alpha \in \{0.02, 0.05, 0.1\}$ for 4.1-4.3 respectively. | $p_i \neq q_i, \mathbb{E}[p_i] \approx \mathbb{E}[q_i]$ | Cont. |
| 5 | $\text{round}(x)$ | Values are binarized. | $p_i \neq q_i$ | Cont. |
| 6.1-6.3 | $b(1-x) + (1-b)x$ $b \sim \text{Bernoulli}(\rho)$ | Values are negated with probability $\rho \in \{0.2, 0.4, 0.6\}$ for 6.1-6.3 respectively. | $p_i \neq q_i,$ $\mathbb{E}[q_i] = \rho + (1 - 2\rho)\mathbb{E}[p_i]$ | Cat. |
| 7 | $\text{MLP}(x)$ | Forward through an MLP with min-max normalization or binarization. | $p_i \neq q_i$ | Both |
| 8 | $P X_i$ | $P$ is a random permutation matrix applied to all features simultaneously. | $p_i = q_i, p_C = q_C, p \neq q$ | Both |
| 9 | $\text{KNN}(x)$ | Predict feature with KNN (Regressor). | - | Cont. |
| 10 | $\text{KNN}(x)$ | Predict feature with KNN (Classifier). | - | Cat. |

**Probabilistic Simulations.** We generate 15 datasets with 1000 features and 5000 samples, simulated from probabilistic distributions including multivariate Gaussians, multivariate Bernoulli distributions, Gaussian mixture models, and Bernoulli mixture models. By having full access to the generating distributions, the real distribution shift and divergence between distributions can be measured. The manipulations include marginal shifts in the mean and/or variance between distributions and shifts in

the feature correlations. Note that here we do not apply the shifts described in Table 1, but instead directly simulate datasets from distributions having a shift. See Section B.2 for more details.

**Experimental Details.** In both shift localization and correction, each method only has access to the reference dataset $X$ and the corrupted query dataset $Y$, while ground truth information, such as actual corrupted feature locations $C^*$ or the original (pre-shifted) query dataset $Y^*$, is not accessible. The localization task is evaluated by comparing the localized corrupted features $C$ and the true locations $C^*$ with the F-1 score, and the correction task is evaluated by comparing the corrected query dataset $Y'$ and the reference dataset with non-parametric empirical divergence estimators. We use the simulated datasets to perform hyperparameter search for DataFix and all competing methods, while the real datasets are used as a hold-out testing set (see below and Section G for more details).

**Feature Shift Localization.** We evaluate our method and 8 competing techniques, including four feature shift localization methods (MB-SM, MB-KS, KNN-KS, and Deep-SM) and four feature selection methods (MI, selectKbest, MRMR, and Fast-CMIM) (Fig. 3). We evaluate MB-SM, MB-KS, KNN-KS, and Deep-SM with both their recommended configuration, without a priory specification of the number of corrupted features $|C|$, and with the hyperparameter configuration that yielded optimal results and includes the ground truth $|C|$ (with *). The rest of the methods do not have access to the ground truth $|C|$. We measure the F-1 score of feature shift localization and average it across the percentage of manipulated features and manipulation types. Figure 3 (left) shows the median and mean F-1 scores across the real and simulated datasets, respectively. We make use of the median because some F-1 scores are missing as some techniques can not process the larger datasets such as "Founders" and "Canine" given the assigned time budget of 30h. DataFix outperforms all the competing techniques, in both real and simulated datasets, even when compared with MB-SM, MB-KS, KNN-KS, and Deep-SM that make use of the ground truth $|C|$.

Figure 3 (right) shows the median (left) and mean (right) F-1 scores divided by type of manipulation and datasets. DataFix outperforms all competing methods in all types of manipulations, except for type 10, where selectKbest provides a higher F-1 score. Techniques using univariate tests, MRMR and Fast-CMIM provide good results for manipulations causing marginal distribution shifts, but completely fail when facing manipulations affecting feature correlations (manipulations 3 and 8), while conditional testing-based techniques (MB-SM, MB-KS, KNN-KS, and Deep-SM) and DataFix are able to detect them. DataFix outperforms competing methods in most of the real datasets, with an overall lower F-1 score for datasets with a larger number of dimensions (phenotypes, founders, and canine).

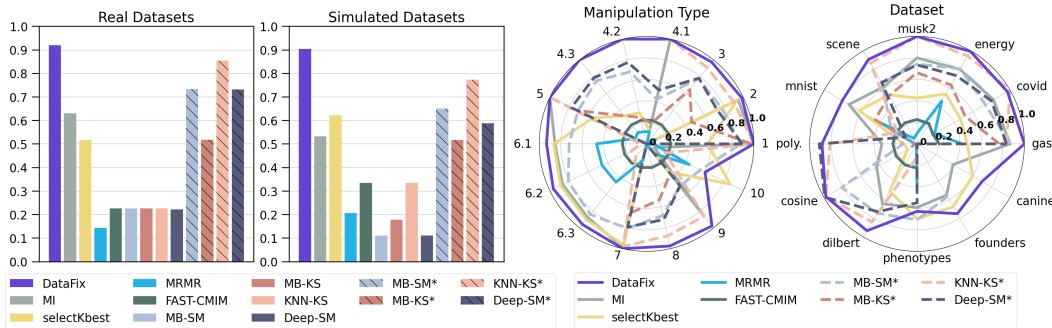

Figure 3: (left) median and mean F-1 score of real and simulated datasets, and (right) median and mean F-1 score across manipulation types and datasets. Higher is better.

**Feature Shift Correction.** We evaluate our method and 13 competing techniques, including predictive models like KNN, linear regression (LR), and multilayer perceptron (MLP), imputation methods including HyperImpute, ICE, MIRACLE, MissForest, and SoftImpute, optimal transport methods such as Sinkhorn, adversarial learning methods like GAIN, and domain adaptation techniques including deep destructors (DD) and iterative alignment flows (INB). We perform a manipulation-agnostic evaluation, where we set the corrupted features to 0 (or missing) and treat all manipulation types equally. Both the reference and the query are provided to each method, and non-parametric statistical metrics computed between the reference and the updated (corrected) queries are used to evaluate each method's performance. Specifically, we use the Wasserstein distance ($W_2^2$) as in [48], the empirical estimator of the Henze-Penrose divergence ($D_{hp}$) [85, 86, 87], and the empirical

estimator of the symmetric Kullback-Leiber divergence ($D_{skl}$) [88]. We report the metrics after subtracting the "background" divergences computed with the reference and query datasets previous to any manipulation. Figure 4 shows the median and mean metrics for the real and simulated datasets respectively. DataFix is able to provide a corrected query dataset $Y'$ with the lowest empirical divergences. Despite their simplicity, KNN and linear regression provide competitive results, followed by MLP, HyperImpute, ICE, INB, and Sinkhorn. See Section J for more details.

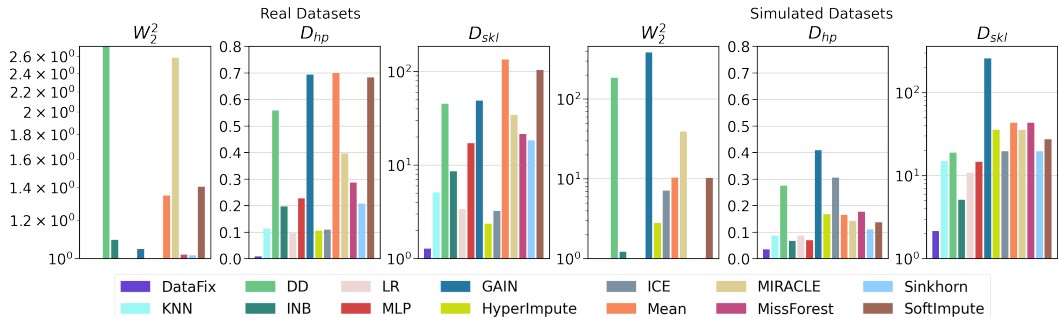

Figure 4: $W_2^2$, $D_{hp}$, and $D_{skl}$ of real and simulated datasets. Lower is better.

**DataFix Analysis.** Figure 5 shows the iterative process of DF-Locate before and after performing shift correction, in simulated dataset 12 with 200 corrupted features out of 1000 (see Section B.2 for more details). Fig. 5 (left) shows the total variation distance estimated by the random forest (blue) which lower-bounds its ground truth monte carlo estimate (black) as corrupted features are detected and removed. The F-1 detection score increases until all features are detected and the iterative process is stopped. Fig. 5 (right) shows the iterative process applied to the corrected query with different methods. ICE and DD provide an updated query that leads to a lower empirical divergence, while the other methods provide an updated query that increased the shift instead of reducing it. DF-Correct (Purple) provides an accurately corrected query, with no empirical divergence detected by DF-Locate.

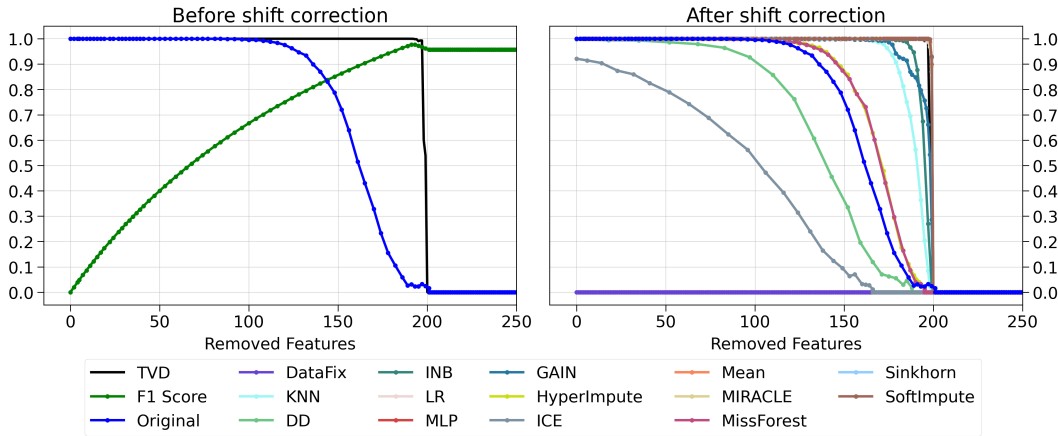

Figure 5: DF-Locate iterative process before and after shift correction.

Extended experimental results are present in the appendix, including an analysis of the method's computational time (Section H), the effect of the classifiers used in DataFix (Section K), experimental results of using corrected datasets in downstream classification and regression (Section M), a detailed division of the quantitative results (Sections I and J), and a discussion of limitations (Section O).

## 7 Conclusions

In this paper, we introduced a new framework "DataFix" which makes use of tree-based classifiers, combined with iterative heuristics, to localize and correct feature shifts. The system, inspired by adversarial learning and feature selection frameworks, is able to accurately detect and correct a wide range of distribution shifts in many types of datasets, surpassing existing techniques.

## Acknowledgments and Disclosure of Funding

This work was partially supported by a grant from the Stanford Institute for Human-Centered Artificial Intelligence (HAI) and by NIH under award R01HG010140. This research has been conducted using the UK Biobank Resource under Application Number 24983.

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

# A  Extended Related Work

The task of feature shift localization and correction is connected to various areas of statistics and machine learning, including out-of-distribution (OOD) generalization, outlier detection, and Generative Adversarial Networks (GANs), among many others.

**Out-of-distribution Generalization**. Similar to feature shift localization and correction, OOD generalization addresses the issue of divergence between datasets or data sources. It does so by aiming to enhance the test (target) accuracy in settings where the test data significantly deviates from the training (source) data distribution. This divergence is often a consequence of the target domain being distinct from the source domain used for training the machine learning models. While both shift localization/correction and OOD generalization aim to reduce distribution shifts, OOD generalization has a primary focus on the task of domain adaptation, setting it apart from the primary objectives of feature shift localization and correction. Extensive work on OOD generalization includes studies into the conditions under which a classifier trained on source data can be expected to perform effectively on target data [89], learning invariant representations for domain adaptation [90], generalizing to unseen domains via distribution matching [91], and using neural networks to learn representations that possess discriminative qualities for the main learning task while remaining indiscriminate concerning the domain shift [92]. The latter goal partially aligns with the objective of feature shift localization and correction, excluding conditional shift, where class-conditional distributions of input features change between source and target domains.

**Anomaly Detection**. Anomaly detection finds observations that deviate noticeably from the data distribution. It has widespread applications in fault detection, particularly in industrial processes, where training samples with normal patterns are used to identify operations that deviate from those. The field of anomaly detection has seen extensive research, including machine learning approaches [93] and deep learning techniques [94, 95, 96], particularly auto-encoders [91, 1].

**Generative Adversarial Networks**. GANs have gained extensive traction in the imputation of missing data, facilitating the creation of realistic fake data through adversarial training. Among these, GAIN [49] stands out as one of the most renowned works, serving as the foundation for subsequent advancements. These improvements include data augmentation, a feature enhanced by MisGAN [97], architectural and loss modifications exemplified by GAMIN [98], SGAIN, WSGAIN-CP, and WSGAIN-GP [99], the incorporation of implicit label information as demonstrated by PC-GAIN [100], the ensemble of GANs by MI-GAN [101] and the capability to combine global and local information demonstrated by GAGIN [102].

# B Datasets

## B.1 Real Datasets

In this work, we use a total of 9 continuous and 3 categorical datasets, with their dimensions and data types shown in Table 2. The Gas, Covid, and Energy datasets are the same as those used in [9], and we apply identical preprocessing procedures. The Musk2 dataset provides information on musk and non-musk molecules. The Scene dataset describes image characteristics. The MNIST dataset consists of images depicting handwritten digits. The Dilbert dataset is an image recognition dataset of pictures of objects rotated from various orientations. We additionally include 3 internal datasets of biomedical data: Phenotypes, Canine, and Founders datasets. The Phenotypes dataset contains a subset of categorical phenotypes from the UK Biobank, following the same pre-processing as in [103]. The Canine and Founders datasets comprise binary-coded sequences of DNA including Single Nucleotide Polymorphisms (SNPs), representing data for multiple dog breeds and human populations, respectively.

The Polynomial and Cosine are two internally generated datasets consisting of values obtained from deterministic simulations. The Polynomial dataset includes samples where each feature value is obtained by evaluating a second-degree polynomial function $f(x) = ax^2 + bx + c$. The parameters $a$, $b$, and $c$ are fixed for each sample. The feature values are derived by evaluating the polynomial function for the different $x$ values within a sample. The values for $x$, $a$, $b$, and $c$ are uniformly sampled from the range $[-10, 10]$. The Cosine dataset consists of samples with features following a cosine function $f(x) = a \cdot \cos(bx + c)$. Similar to the Polynomial dataset, $a$, $b$, and $c$ are fixed for each sample, and each feature value is obtained by evaluating the cosine function for the different $x$ values within a sample. Here, the values of $b$ and $c$ are uniformly sampled within the range $[-\pi, \pi]$.

Table 2: Datasets used to evaluate DataFix.

| Dataset | No. of attributes | No. of samples | Data |
|---|---|---|---|
| Gas | 8 | 12,815 | Cont. |
| Covid | 10 | 9,889 | Cont. |
| Energy | 26 | 19,735 | Cont. |
| Musk2 | 166 | 6,598 | Cont. |
| Scene | 294 | 2,407 | Cont. |
| MNIST | 784 | 70,000 | Cont. |
| Phenotypes | 1,227 | 31,424 | Cat. |
| Dilbert | 2,000 | 10,000 | Cont. |
| Founders | 10,000 | 4,144 | Cat. |
| Canine | 198,473 | 1,444 | Cat. |
| Cosine | 1,000 | 10,000 | Cont. |
| Polynomial | 1,000 | 10,000 | Cont. |

## B.2 Simulated Probabilistic Datasets

We generate 15 simulated datasets containing 1000 features and 5000 samples by sampling from pre-defined probabilistic distributions, including multivariate Gaussians, with and without transformations, multivariate Bernoulli distributions, Gaussian mixture models, and Bernoulli mixture models. A total of 200 features are shifted in each dataset such that $|C| = 200$ and $|\overline{C}| = 800$. Table 3 describes the distributions used in each dataset. For every dataset, two distributions, $p$ and $q$, are defined, so that $D(p, q) > 0$ but $D(p_{\overline{C}}, q_{\overline{C}}) = 0$. In fact, for all datasets except for dataset 8, we also have that $D(p_C, q_C) > 0$. A further discussion of the effect of shifts with $D(p_C, q_C) > 0$ and with $D(p_C, q_C) = 0$, and its relation to the equivalence of feature shift localization and feature selection, is provided in the following sections. Because we have access to $p$ and $q$, we can compute the real divergence between the distributions. In practice, we make use of a Monte Carlo estimate, as shown in Figures 5 and 25, because the divergences might not have a closed-form solution or can be computationally intractable.

Datasets 1-3 are based on a multivariate Gaussian, with diagonal covariance used in datasets 1 and 2, and a covariance $\Sigma$ used in dataset 3. The covariance matrix $\Sigma$ is defined by a Gaussian kernel such that the $ij$ component is $\Sigma_{ij} = \exp\frac{-||i-j||^2}{s}$, where $s$ acts as a scale parameter, and $0 \leq \Sigma_{ij} \leq 1$ with $\Sigma_{ii} = 1$. In practice, when constructing the covariance matrix, we perform a shuffle of the feature order to better depict tabular data, where, in many cases, the correlation between features does not follow any specific ordering (opposed to images or audio). We use $s = 0.05$ to define $\Sigma_{ij}$ in dataset 3. Datasets 4 and 5 follow a lognormal distribution, defined as $X = \exp(V)$ with $V \sim \mathcal{N}(\mu, \Sigma)$ and $X \sim \text{Lognormal}(\mu, \Sigma)$. We use $s = 0.05$ and $s = 0.002$ to define $\Sigma_{ij}$ in datasets 4 and 5, respectively. Datasets 6-8 follow a logit-normal distribution defined as $X = \sigma(V)$ with $V \sim \mathcal{N}(\mu, \Sigma)$ and $X \sim P(\mathcal{N}(\mu, \Sigma))$, where $\sigma$ is the sigmoid transformation. We use $s = 0.05$, $s = 0.002$, and $s = 0.002$ to define $\Sigma_{ij}$ in datasets 6, 7, and 8, respectively. Datasets 9-12 follow a multivariate Bernoulli with independent features. Each feature $i$ has a frequency of $f_i$, where $f \sim P(\mathcal{N}(0, 2I))$, $\epsilon \sim \mathcal{N}(0, I)$, and $(\cdot)_{0,1} = \text{clamp}_{0,1}(\cdot) = \min(\max(\cdot, 0), 1)$ is the clamping function to ensure that the frequencies are between 0 and 1. Dataset 13 follows a Gaussian Mixture Model distribution, with 3 mixtures of equal weights, $\mu_i \sim \mathcal{N}(0, 0.01I)$, and $\Sigma_i$ defined with $s = 0.3$. Datasets 14 and 15 follow a Bernoulli Mixture Model distribution with 3 mixtures and $f_i \sim \text{Uniform}^d(0, 1)$.

Datasets 1, 3, 4, 6, 9, 10, 11, 12, 13, and 15 apply a shift to the marginal means, such that for all $i \in C$, $\mathbb{E}[p_{C_i}] \neq \mathbb{E}[q_{C_i}]$. Such datasets include marginal shifts of a similar nature as the shifts generated by manipulation types 1, 2, and 6 applied to real datasets. Dataset 2 performs a shift of the marginal standard deviation while maintaining their mean such that for all $i \in C$, $\mathbb{E}[p_{C_i}] = \mathbb{E}[q_{C_i}]$ but $p_{C_i} \neq q_{C_i}$ and $\text{Var}[p_{C_i}] \neq \text{Var}[q_{C_i}]$, leading to a marginal shift similar to the one applied by manipulation type 4 used in the real datasets. Datasets 13 and 15 apply a shift to the mean of just one mixture of the mixture model, leading to only $1/3$ of the samples being shifted, while still ensuring that $\mathbb{E}[p_{C_i}] \neq \mathbb{E}[q_{C_i}]$. Datasets 5, 7, and 14 apply a distribution shift consisting of removing the correlation between features, equivalently to manipulation type 3 applied in real datasets, such that $q_C = \prod_{i \in C} q_i$ and $p_C \neq q_C$. Dataset 8 also applies a shift originating from modifying the correlation between features, but in this case, $\Sigma'$ is defined as:

$$\Sigma'_{ij} = \begin{cases} \Sigma_{ij} & \text{if } i, j \in C, \text{ or } i, j \in \overline{C} \\ 0 & \text{if } i \in C \text{ with } j \in \overline{C}, \text{ or } j \in C \text{ with } i \in \overline{C} \end{cases} \tag{9}$$

where the correlation of the features within $C$ and within $\overline{C}$ are maintained, but the cross-correlations between the $C$ and $\overline{C}$ are lost, which leads to a shift equivalent to manipulation type 8 applied in real datasets, such that $p_C = q_C$, $p_{\overline{C}} = q_{\overline{C}}$, but $p \neq q$.

Table 3: Probabilistic datasets used to evaluate DataFix.

| ID | $p_C$ | $q_C$ | Description |
|---|---|---|---|
| 1 | $\mathcal{N}(0, I)$ | $\mathcal{N}(0.5I, I)$ | Multivariate Gaussians with diagonal co-variance and a shifted mean. |
| 2 | $\mathcal{N}(0, I)$ | $\mathcal{N}(0, 1.5I)$ | Multivariate Gaussians with diagonal co-variance and a shifted scale. |
| 3 | $\mathcal{N}(0, \Sigma)$ | $\mathcal{N}(0.5I, \Sigma)$ | Multivariate Gaussians with non-diagonal covariance and a shifted mean. |
| 4 | Lognormal$(0, \Sigma)$ | Lognormal$(0.5I, \Sigma)$ | Multivariate lognormal with non-diagonal covariance and a shifted mean. |
| 5 | Lognormal$(0, \Sigma)$ | Lognormal$(0, I)$ | Multivariate lognormal with non-diagonal $(p_C)$ and diagonal $(q_C)$ covariance. |
| 6 | $P(\mathcal{N}(0, \Sigma))$ | $P(\mathcal{N}(0.5I, \Sigma))$ | Multivariate logit-normal with non-diagonal covariance and a shifted mean. |
| 7 | $P(\mathcal{N}(0, \Sigma))$ | $P(\mathcal{N}(0, I))$ | Multivariate logit-normal with non-diagonal $(p_C)$ and diagonal $(q_C)$ covariance. |
| 8 | $P(\mathcal{N}(0, \Sigma))$ | $P(\mathcal{N}(0, \Sigma'))$ | Multivariate logit-normal with different non-diagonal covariances. |
| 9 | Bernoulli$(f)$ | Bernoulli$((f + 0.05\epsilon)_{0,1})$ | Multivariate independent Bernoulli with a shifted mean. |
| 10 | Bernoulli$(f)$ | Bernoulli$((f + 0.1\epsilon)_{0,1})$ | Multivariate independent Bernoulli with a shifted mean. |
| 11 | Bernoulli$(f)$ | Bernoulli$((f + 0.5\epsilon)_{0,1})$ | Multivariate independent Bernoulli with a shifted mean. |
| 12 | Bernoulli$(f)$ | Bernoulli$((f + 1.0\epsilon)_{0,1})$ | Multivariate independent Bernoulli with a shifted mean. |
| 13 | $\frac{1}{3}\sum_{i=1}^{3}\mathcal{N}(\mu_i, \Sigma_i)$ | $\frac{1}{3}\sum_{i=1}^{3}\mathcal{N}(\mu_i', \Sigma_i)$ | Gaussian Mixture Model with one mixture shifted such that $\mu_1' = \mu_1 + 10$, $\mu_2' = \mu_2$, and $\mu_3' = \mu_3$. |
| 14 | BMM$([f_1, f_2, f_3])$ | BMM$([f', f', f'])$ | Bernoulli Mixture Model with different means $f' = \frac{f_1 + f_2 + f_3}{3}$. |
| 15 | BMM$([f_1, f_2, f_3])$ | BMM$([(f_1 + 0.2)_{0,1}, f_2, f_3])$ | Bernoulli Mixture Model with one mixture shifted. |

## C Feature Selection and Feature Shift Localization Equivalence

DF-Locate combines feature importance and feature selection techniques to localize the feature subset originating the distribution shift. Feature selection techniques often solve, either implicitly or explicitly, the following problem:

$$C = \underset{|C| \leq k}{\operatorname{argmax}} I(z_C; t) \tag{10}$$

where $C$ is the feature subset of size up to $k$, which maximizes the mutual information $I$ between the input sample $z$, restricted to the features in $C$, and the label $t$. In practice, the distributions of $z$ or $t$ are unknown, making Eq. 2 intractable. Instead, most methods predict a score that approximates the amount of information relative to the label present at each feature: $\beta = F(Z, T)$, where $\beta \in \mathbb{R}^d$ and $F(\cdot)$ is a function mapping samples $Z$ and labels $T$ to an importance score for each feature. Informally, these scores provide an ordering such that if features $i$ and $j$ have $|\beta_i| > |\beta_j|$, then $I(z_{\bar{i}}; t) \lessgtr I(z_{\bar{j}}; t)$.

Section 4 shows that by defining $z$ as the samples coming from $X$ and $Y$, and $t$ as the binary label of origin, such that $z|_{t=0} \sim p$ and $z|_{t=1} \sim q$, then Fano's inequality [71] and LeCam's method [72] can be used to relate the mutual information between $z$ and $t$ and the total variation distance between $p$ and $q$. Namely, when the number of manipulated features is known $|C| = k$, and the divergence of the corrupted features is larger than 0, such that $D_{TV}(p, q) = D_{TV}(p_C, q_C) = 1$, and $D_{TV}(p_{\overline{C}}, q_{\overline{C}}) = 0$, then feature selection and feature shift localization are equivalent:

$$C = \operatorname*{argmax}_{|C| \leq k} I(z_C; t) = \operatorname*{argmin}_{D_{TV}(p_{\overline{C}}, q_{\overline{C}})=0} |C| \qquad (11)$$

Even if $|C|$ is unknown, if only marginal shifts are present, one can perform feature shift detection by iteratively solving Eq.2 with $k = 1$, and removing the detected features at each iteration from $z$ as long as $I(z_C; t) > 0$, making the iterative feature selection and feature shift localization equivalent problems:

$$C_i = \operatorname*{argmax}_{|C|=1} I(z_C; t) \qquad (12)$$

where $C = \bigcup_{i=1}^{l} C_i$. In fact, the approach presented in section 4, DF-Locate, can be seen as an approximation of this iterative process, where one or more features are selected at each step by the feature removal policy function.

However, the equivalence of both tasks breaks down for distribution shifts such as the one applied in the manipulation type 8 for real datasets, and in dataset 8 of probabilistic simulations, where $p_C = q_C$ and $p_{\overline{C}} = q_{\overline{C}}$, but $p \neq q$. That is, when considering only the corrupted features $C$ or the non-corrupted features $\overline{C}$ in isolation, the shift is impossible to detect unless all features are considered jointly. Therefore, any feature selection technique that approximates either implicitly or explicitly equation 2, will need a subset of features $G$ containing features from both $C$ and $\overline{C}$ in order to obtain $I(z_G; t) > 0$ because $I(z_C; t) = 0$ and $I(z_{\overline{C}}; t) = 0$. Furthermore, if $|C| = |\overline{C}|$ the problem of feature shift detection becomes unsolvable. This is because, even if a technique was able to properly identify a subset of features $G = C$, it would not be possible to know if the detected subset $G$ contains corrupted or non-corrupted features, that is if $G = C$ or $G = \overline{C}$, making the assumption of $|C| < |\overline{C}|$ a necessary condition.

Figure 3 (right) and Figure 17 show the F-1 score for each manipulation type, indicating that DataFix is able to correctly localize manipulated features with manipulation type 8 on real datasets, despite breaking the equivalence between feature selection and feature shift localization approaches. In contrast, Figure 18, which illustrates the average F-1 on the probabilistic datasets, shows that dataset 8 is the one providing the lowest F-1 scores. While the low F-1 score in dataset 8 can be partly caused by the mismatch between feature selection and feature shift localization problems, it can also originate from the difficulty of detecting shifts caused by mismatching correlations, as it provides similar performance as in datasets 5 and 7, where correlation shifts (with $p_C \neq q_C$) are applied.

# D   Feature Shift from Imputation Methods

Imputation and supervised methods trained to reduce the expected mean square error (MSE) between the predicted $\hat{x}_C$ and real $x_C$ features of a given sample $x$ can lead to distribution shifts. Note that the optimal function minimizing $\mathbb{E}_{x \sim P}[||x_C - f(x_{\overline{C}})||^2]$ is the expected value of $x_C$ conditioned in $x_{\overline{C}}$, that is $f^*(x_{\overline{C}}) = \mathbb{E}[x_C | x_{\overline{C}}]$. Therefore, the method that predicts the corrupted (or missing) features $C$ given the non-corrupted (non-missing) features $\overline{C}$, which provides the lowest MSE, will generate predicted samples $\hat{x}_C = f^*(x_{\overline{C}})$, with a distribution where $\mathbb{P}(\hat{x}_C) = 1$ for $\hat{x}_C = \mathbb{E}[x_C | x_{\overline{C}}]$, and $\mathbb{P}(\hat{x}_C) = 0$ everywhere else. If $\mathrm{Var}[x_C | x_{\overline{C}}] > 0$, then $D(\mathbb{P}(\hat{x}_C), \mathbb{P}(x_C)) > 0$ because $\mathrm{Var}[\hat{x}_C | x_{\overline{C}}] = 0$.

For example, given a dataset of samples $x_1, x_2, ..., x_N$, with $x_{i\overline{C}} = x_{j\overline{C}}$ and $x_{iC} \neq x_{jC}$ for all $i$ and $j$, in other words, $\mathrm{Var}[x_C | x_{\overline{C}}] > 0$, the optimal regression model, in terms of MSE, will predict $\hat{x}_{iC} = \mathbb{E}[x_{iC} | x_{i\overline{C}}]$ for all $i$, such that $\mathrm{Var}[\hat{x}_C | x_{\overline{C}}] = 0$. Similarly, consider a dataset where the features $C$ and $\overline{C}$ are independent, such that $p = p_C p_{\overline{C}}$, and $p_C = \mathcal{N}(\mu, I)$, then $\hat{x}_C = \mathbb{E}[x_C | x_{\overline{C}}] = \mathbb{E}[x_C] = \mu$, where $\mu \in \mathbb{R}^{|C|}$ is a constant vector. The simulated probabilistic dataset 1 follows this form, with $p_C = \mathcal{N}(0, I)$ and $q_C = \mathcal{N}(0.5I, I)$, where the method providing the lowest MSE will be the one predicting $\hat{x}_C = \mathbb{E}[x_C | x_{\overline{C}}] = \mu = 0$ for all samples. Figure 6 shows the histogram of the first feature values before and after performing feature correction with multiple methods. Most techniques, especially imputation-based techniques, output the mean or values highly close to it, which while producing minimum MSE, does not reflect the real distribution, removing or reducing its variance and leading to a distribution shift.

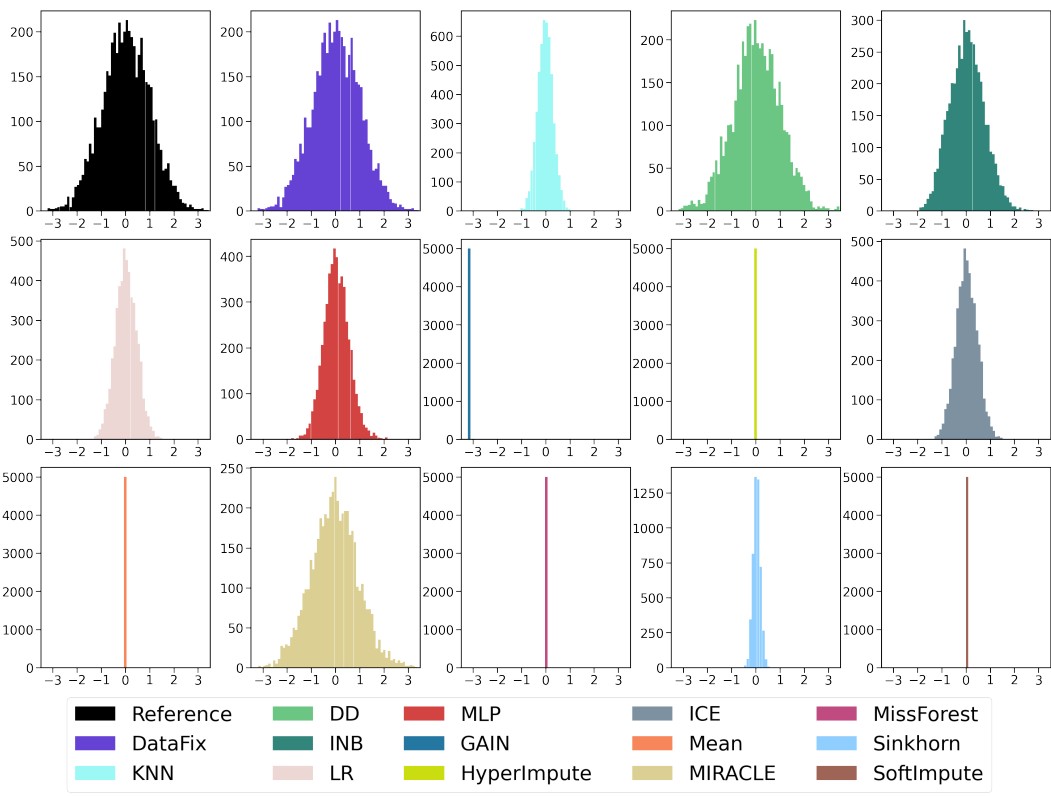

Figure 6: Histograms of first feature values before and after shift correction with various techniques for probabilistic dataset 1.

# E   DF-Locate

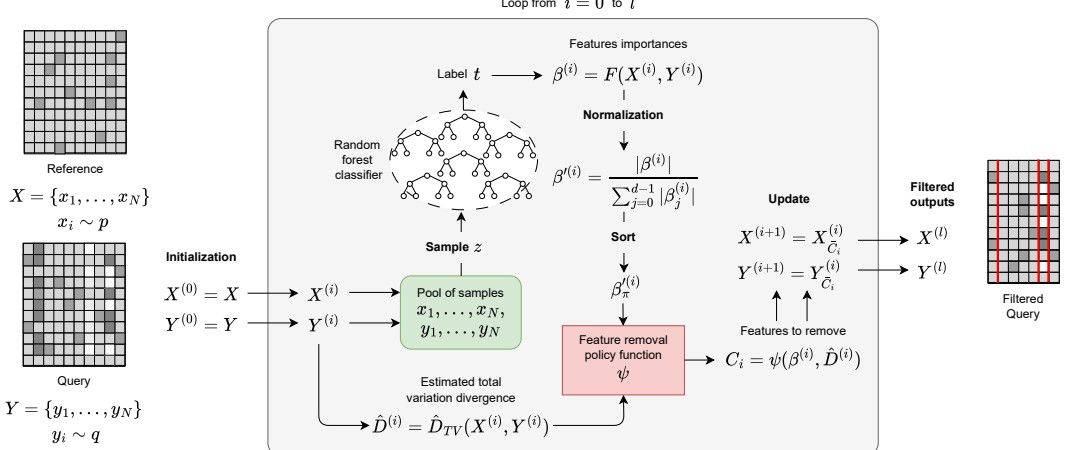

Figure 7: DF-Locate diagram.

DF-Locate (Section 4, Figures 1 and 7, and Algorithm 1) is the proposed method within DataFix that localizes the features originating the distribution shift by performing feature selection in an iterative way. First, starting with $i = 0$, and a reference $X^{(0)} = X$ and query $Y^{(0)} = Y$ datasets, a set of discriminators are trained $\theta = \operatorname{argmax}_\theta \hat{D}_\theta(X^{(i)}, Y^{(i)})$. The discriminators are used to predict the empirical total variation distance (TVD) between distributions $\hat{D} = \hat{D}_\theta(X^{(i)}, Y^{(i)})$, and a feature importance score for each feature $\beta = F_\theta(X^{(i)}, Y^{(i)})$. The divergence and feature importances are used to select potentially corrupted features with the feature removal policy function (see section 4) $C_i = \psi_\tau(\beta, \hat{D})$. Then, the detected features are removed from $X$ and $Y$, such that $X^{(i+1)} = X^{(i)}_{\bar{C}_i}$ and $Y^{(i+1)} = Y^{(i)}_{\bar{C}_i}$. The process is repeated as long as the estimated divergence is smaller than a threshold $\hat{D}_\theta(X^{(i)}, Y^{(i)}) > \epsilon$, less than half of the features of the dataset are removed $k^{(i)} < \frac{|Y|}{2}$, or at least one feature is selected by the feature removal policy function at each step $k^{(i)} - k^{(i-1)} > 0$. In this work we use $\epsilon = 0.02$ as the stopping threshold. After the iterative process is stopped, a refinement step, described below, is applied to remove features that might have been incorrectly selected as corrupted.

## E.1   Shift Detection

We detect whether a shift is present by estimating an $f$-divergence between the distributions $p$ and $q$. Specifically, we make use of the variational form [67, 68, 69]:

$$D_f(p, q) \geq \mathbb{E}_{x \sim p}[f'(r_\theta(x))] - \mathbb{E}_{y \sim q}[f^*(f'(r_\theta(y)))] \tag{13}$$

where $r_\theta(x)$ is a function approximating the likelihood ratio between $p$ and $q$. The inequality becomes an equality when $r_\theta(x) = r^*(x) = p(x)/q(x)$ is the true likelihood ratio function, defining the Bayes decision rule that optimally separates $p$ and $q$. $f$ is the generator function of the $f$-divergence, $f'$ is its first order derivative and $f^*$ is its Fenchel conjugate [70].

In practice, estimating $D_f(p, q)$ is intractable, as we do not have direct access to $p$ and $q$. Hence, we use an empirical estimator of the divergence by training a binary classifier $\mathcal{D}_\theta(x)$ to discriminate the samples coming from $X = \{x_1, ..., x_N\}$ and $Y = \{y_1, ..., y_N\}$, with $x_i \sim p$ and $y_j \sim q$. The likelihood ratio can be easily estimated as $r_\theta(x) = \mathcal{D}_\theta(x)/1-\mathcal{D}_\theta(x) = \exp \sigma^{-1}(\mathcal{D}_\theta(x))$, where $\sigma^{-1}$ is the inverse sigmoid. We use $k$-fold train-evaluation split, where the dataset is divided into $k = 5$ subsets. At each iteration, 80% of the samples are used to train a random forest classifier $\mathcal{D}_\theta(x)$, and the 20% left is used to estimate the empirical divergence:

$$\hat{D}_\theta(X, Y) = \frac{1}{N_x} \sum_{i=1}^{N_x} f'(r_\theta(x_i)) - \frac{1}{N_y} \sum_{j=1}^{N_y} f^*(f'(r_\theta(y_j))) \tag{14}$$

where $N_x$ and $N_y$ are the number of testing samples of the reference and query datasets respectively, and $D_f(p, q) \geq \mathbb{E}_{x \sim p, y \sim q}[\hat{D}_\theta(X, Y)]$. By training the random forest, we are approximately finding the discriminator that provides a maximal divergence $\theta^* = \arg\max_\theta \hat{D}_\theta(X, Y)$ within the random forest hypothesis space.

## E.2 Feature Removal Policy Function

The feature removal policy function $\psi$, introduced in section 4, makes use of the predicted feature importances $\beta = F_\theta(Z, T)$ and the estimated total variation distance $\hat{D} = \hat{D}_{TV}(X, Y)$ to select the corrupted features $C = \psi(\beta, \hat{D})$. This function utilizes the sorted normalized absolute value of the importance scores $\beta'_\pi = \frac{|\beta|_\pi}{\sum_{i=0}^{d-1} |\beta_i|}$ with $\beta'_{\pi(0)} \geq ... \geq \beta'_{\pi(d-1)}$. At each iteration, the first $J$ features are selected as corrupted, where $J$ is dynamically selected at each iteration (equation 4) as the smallest index $k$ such that $\gamma(k) = \sum_{j=0}^{k} \beta'_{\pi(j)} \geq \tau \hat{D}$ (Eq. 4), with $\tau$ selected by cross-validation. The features selected as corrupted $C$ include the features from 0 to $J$ with scores higher than $\frac{1}{d}$ (Eq. 5).

The product $\tau \hat{D}$ specifies a threshold that defines how many features are selected as corrupted at each iteration. When $\tau \hat{D}$ is small, $\psi$ returns the feature with the highest feature importance, while a larger $\tau \hat{D}$ leads to more features flagged as corrupted. The value of $\tau$ is selected through hyperparameter optimization using the simulated datasets and maintained fixed afterward, and $\hat{D}$ is computed at each iteration with values that are, in expectation, non-increasing. When the empirical shift is low, the product $\tau \hat{D}$ is small, leading to a smaller number of features being flagged as corrupted. Figure 8 presents a comparative analysis of the sorted normalized feature importance scores $\beta'_\pi$ in the first, middle, and last iterations of the DF-Locate iterative process at the first, second, and third figure column respectively. Different thresholds $\tau$ (0.05, 0.1, 0.2, and 0.3) are used to localize and filter the corrupted features at each iteration, with each row representing a specific threshold value. The features flagged as corrupted (as part of $C$) by the feature removal policy function $\psi$ are indicated in red color. As the number of iterations increases, $\hat{D}$ decreases and fewer features are flagged as corrupted, with the last iteration containing only a small number of remaining corrupted features and most features exhibiting similar low importance scores. Larger thresholds $\tau$ lead to more features selected at each iteration. However, for thresholds higher than 0.1, some of the selected features can be falsely classified as corrupted, indicating the need for lower threshold values.

Figure 9 shows the number of removed features at each iteration with different threshold values $\tau$. Across all threshold values, the number of removed features decreases as the number of iterations increases, and in turn, the empirical divergence decreases. Note that higher threshold values lead to more features removed at each iteration, resulting in a smaller total number of iterations and faster computational times. However, higher thresholds also lead to an increased number of false positives, where many non-corrupted features are incorrectly selected as corrupted, leading to low F-1 scores. Therefore, the $\tau$ threshold provides a trade-off mechanism between feature shift localization speed and accuracy.

## E.3 Refinement Stage

In order to perform the refinement step, we store all intermediate steps so that we can revisit the iterative filtering process and select the optimal stopping point. At each iteration, we store the indexes of the features selected as corrupted, their corresponding estimated $\hat{D}$, and the number of removed features. While the ideal stopping iteration would be the iteration providing the highest F-1 score, this requires access to the ground truth, which is unavailable in practice. However, there exists a point at which the cost of removing additional features, some potentially non-corrupted, outweighs the decrease in the empirical divergence. To determine the optimal iteration, we locate the elbow or knee from a processed curve depicting the empirical total variation distance (TVD) as a function

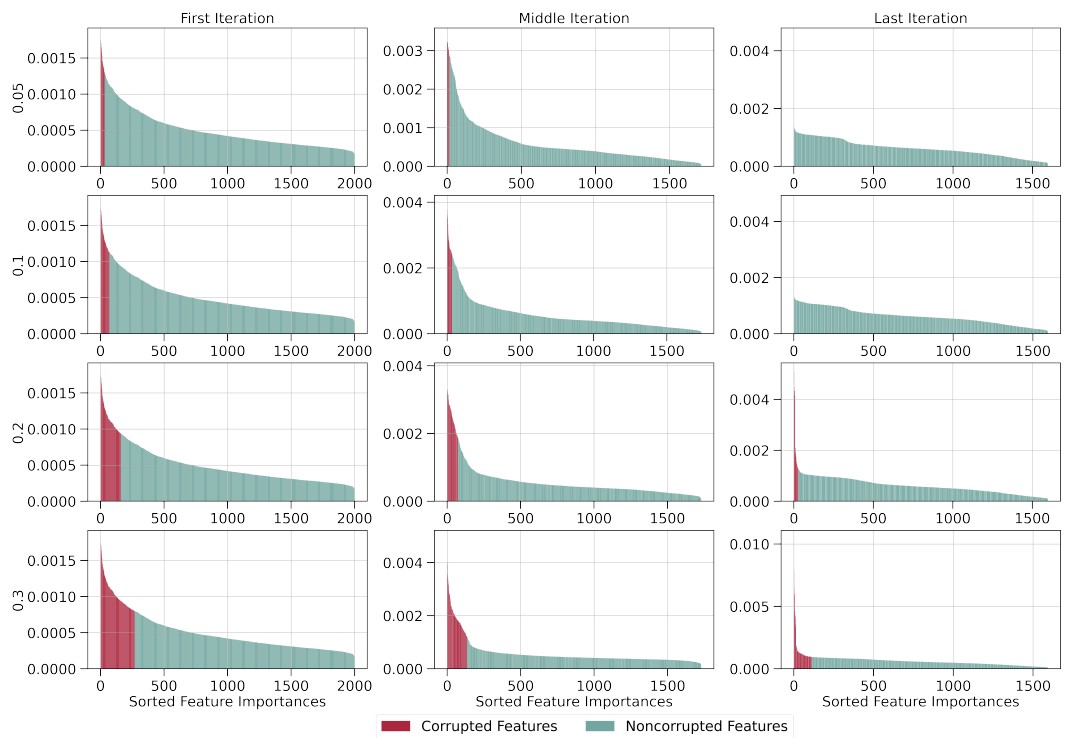

Figure 8: Sorted and normalized feature importance scores of the discriminators in the first, middle, and last iterations of DF-Locate applied to the Dilbert dataset with manipulation type 8. Each row represents a different threshold used to localize and filter the corrupted features at each iteration.

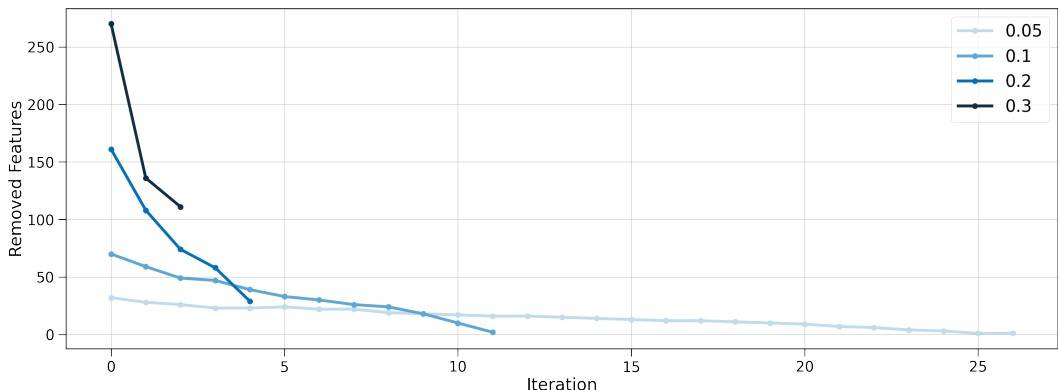

Figure 9: Number of features removed at each iteration for different threshold $\tau$ values in DF-Locate applied to the Dilbert dataset with manipulation type 8.

of the total number of removed features (Figure 10). Once such optimal iteration is determined, all selected features up to that iteration are flagged as corrupted. The curve of the true divergence would always exhibit a convex non-increasing shape but due to the intrinsic randomness in the training and evaluation process of the discriminator, this ideal shape is not always achieved, making the task of locating the knee challenging. In order to make the curve smooth and non-increasing, we make use of the Savitzky-Golay filter [104] due to its ability to remove noise without distorting the underlying signal. Additionally, we apply an opening operation to eliminate any local maxima in the curve, ensuring that each point is equal to or smaller than its left neighbor. Furthermore, we process the initial iterations of the curve to enforce a strict decreasing behavior. After processing the empirical divergence curve, we employ the knee locator method introduced by [105] to identify the optimal stopping criterion.

The Savitzky-Golay filter relies on two main parameters: the length of the filter window and the polyorder used for fitting the samples. We define the window length as $\max(5, 2\lfloor \zeta\delta/2\rfloor + 1)$, where $\zeta$ is chosen from the set $\{1, 2, 3, 5, 7\}$ through cross-validation, and $\delta$ represents the average number of removed features at each iteration. We explore different polyorders from the set $\{3, 4\}$. Based on experiments with simulated datasets, we select $\zeta = 2$, and the best polyorder as $4$. The knee locator involves two primary parameters: sensitivity ($S$) and online mode. The sensitivity parameter determines the number of "flat" points we anticipate encountering in the original data curve before identifying a knee, while the online mode enables the correction of previous knee values. We explore different values for $S$ from the set $\{1, 3, 5, 7\}$. Our experimentation reveals that the optimal values are $S = 5$ and using the offline mode.

Figure 10 shows an example of the knee location, used to refine the selection of features. The F-1 score can serve as a way to evaluate the selected stopping point, showing the trade-off between reduced TVD and the number of removed features.

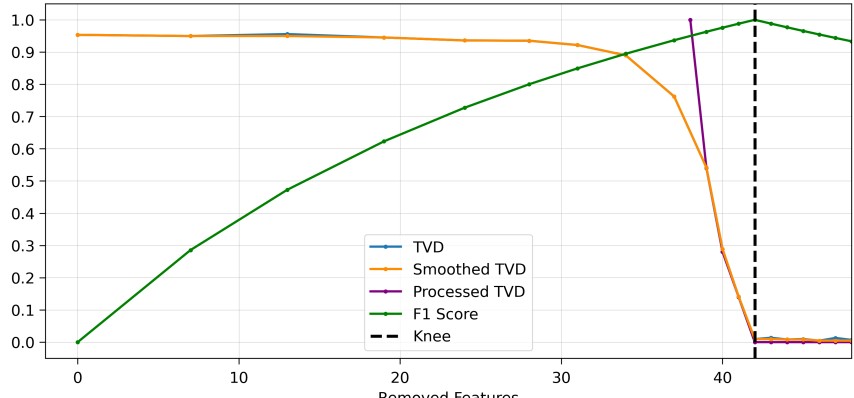

Figure 10: Knee location based on distribution shift. Number of removed Features, TVD, and F-1 score in shift localization task.

# F DF-Correct

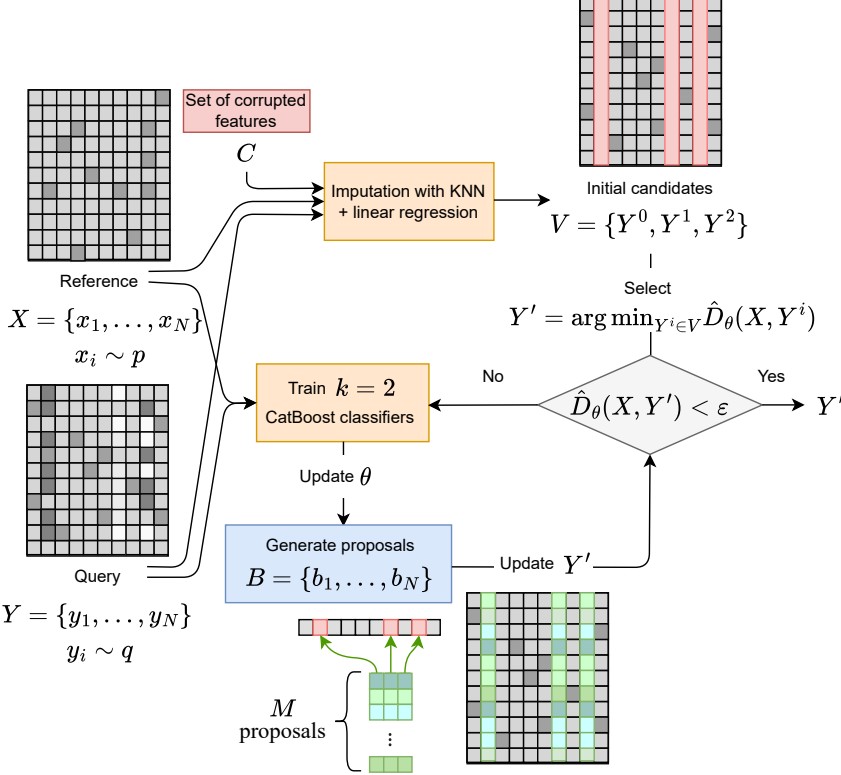

Figure 11: DF-Correct diagram.

DF-Correct (Section 5, Figures 2 and 11, and Algorithm 2) is the proposed method within DataFix that corrects the feature shifts by changing the values of the features in $Y$ originating the distribution shift through an iterative approach. Given the set of corrupted features $C$, DF-Correct tries to generate a new query dataset $Y'$, such that $Y_{\overline{C}} = Y'_{\overline{C}}$, while $D(X, Y') < D(X, Y)$. DF-Correct starts by obtaining an initial candidate of $Y'$ by setting the values within the subset $C$ of $Y$ as missing and performing imputation with linear regression and KNN. Furthermore, a naive initial candidate is generated by replacing the corrupted values of $Y$ with randomly selected values of $X$ (restricted to the features subset $C$). Note that this naive initialization already fixes distribution shifts where features are completely independent of each other. The set of the 3 initial candidates, $V = \{Y^0, Y^1, Y^2\}$, is evaluated by computing the empirical total variation distance with a classifier, and the one providing the lowest empirical divergence is selected as the initial corrected query $Y' = \operatorname{argmin}_{Y^i \in V} \hat{D}_\theta(X, Y^i)$. If $\hat{D}_\theta(X, Y') < \epsilon$, the correction process is finished and $Y'$ is returned. Otherwise, if $\hat{D}_\theta(X, Y') > \epsilon$, an iterative process where some samples of $Y'$ are modified is performed.

The iterative process tries to find new values of $Y' = \{y'_1, y'_2, ..., y'_{N_y}\}$ that reduce the empirical total variation distance:

$$\hat{D}_\theta^{TV}(X, Y') = \frac{1}{N_x} \sum_{i=1}^{N_x} g(r_\theta(x_i)) - \frac{1}{N_y} \sum_{j=1}^{N_y} g(r_\theta(y'_j)) \tag{15}$$

with $g(u) = \frac{1}{2}\text{sign}(u - 1)$. Because the values of $X$ are left untouched, this becomes equivalent to solving:

$$Y' = \underset{Y}{\arg\min} \max_{\theta} \sum_{j=1}^{N_y} -g(r_\theta(y'_j)) = \underset{Y}{\arg\max} \max_{\theta} \sum_{j=1}^{N_y} g(r_\theta(y'_j)) \qquad (16)$$

In order to perform such an optimization process, a set of classifiers are trained using $X$, $Y'$, and data augmentation consisting of performing random permutations within the features of the reference $X$ dataset, in order to generate extra samples of "corrupted" sequences. After training the classifiers, the next step is to find which samples need to be corrected. Note that when $p = q$, we have $\mathbb{E}[\hat{D}_\theta^{TV}(X, Y)] = 0$, and:

$$\mathbb{E}[\sum_{j \in N_y} g(r_\theta(y_j))] = \mathbb{E}[\sum_{m \in N^+} g(r_\theta(y_m))] - \mathbb{E}[\sum_{n \in N^-} g(r_\theta(y_n))] = 0 \qquad (17)$$

where $N^+ = \{i : r_\theta(y_i) > 1\}$ are the indices of samples classified as positive by the discriminator, and $N^- = \{i : r_\theta(y_i) < 1\}$ are the indices of samples classified as negative. Furthermore, both sets have, in expectation, the same size $\mathbb{E}[|N^+|] = \mathbb{E}[|N^-|] = \mathbb{E}[\frac{|N_y|}{2}]$ when $p = q$. This indicates that when both distributions are equal, a discriminator will approximately classify half of the samples as positive and half as negative. Therefore we only correct the set of samples $L$, including up to $\frac{|N_y|}{2}$ samples with the highest probability of being corrupted:

$$L = \{i : r_{\theta_i}(y_i) < r_{\theta_{i+1}}(y_{i+1}) < 1\} \qquad (18)$$

with $|L| \le \frac{|N_y|}{2}$. Next, we construct a set of feature value proposals $B$ that will replace the corrupted features. This proposal set is constructed by including within $B$ all the feature values of the reference $X$, of the initial imputed candidates $V$, of the current corrected query $Y'$, and random permutations of $X$. Then, for every sample $i \in L$, each of the proposals $b \in B$ is placed as an alternative to the corrupted features, generating a sample $y_i^{(b)}$, where $y_i^{(b)}{}_C = b$, and $y_i^{(b)}{}_{\overline{C}} = y_{i\overline{C}}$. The proposal providing the highest probability of being "non-corrupted" is selected:

$$b_i = \underset{b \in B}{\arg\max} \, r_{\theta_i}(y_i^{(b)}) \qquad (19)$$

Finally, the updated sample $y_i^{(b)}$ is placed inside the corrected query $Y'$. After updating all samples in $L$, the divergence is computed again, and if $\hat{D}_\theta(X, Y') > \epsilon$, the process is repeated for a number of epochs. Typically, the number of epochs is set to 1 or 2, as the correction process can become computationally intensive for large datasets (see following sections).

### F.1 MNIST Example Proposals

In Figure 12, we provide a visual comparison of proposals generated from a single iteration of DF-Correct using two image samples from the MNIST dataset, categorized from worst to best based on their likelihood of being corrupted. The comparison is made against their manipulated counterparts with manipulation type 2. The findings reveal that the top proposals generated by DF-Correct excel at correcting the distorted values. The outcomes closely resemble unaltered MNIST images, eliminating any distortions effectively.

### F.2 Relationship between DF-Correct and KNN Imputation

We can provide more insight into how DF-Correct is able to reduce the divergence between datasets by comparing it with KNN imputation. A KNN imputer fills the missing feature values with a combination of the top-k closest samples with respect to L2 distance on non-missing features. Instead, our correction system replaces the corrupted feature values with the top-1 candidate with respect to the probability estimated by the discriminator. In a sense, the KNN training samples are replaced by our feature proposals, and the KNN L2 distance is replaced by the prediction of a discriminator, making the correction task similar to a divergence-reducing imputation problem.

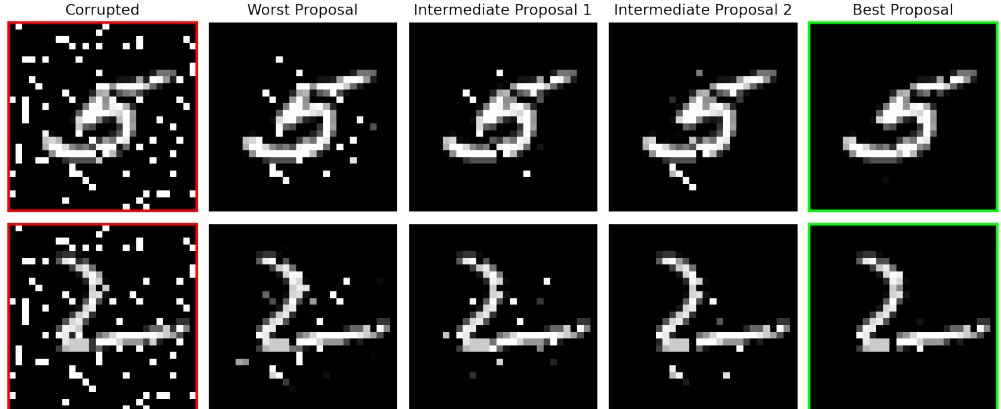

Figure 12: Comparison of worst, intermediate, and best proposals from a single iteration of DF-Correct on two samples of the MNIST dataset against their corrupted version with manipulation type 2.

# G  Experimental Details

## G.1  Hyperparameter Search

In this section, we present the experimental details of our hyperparameter optimization process for DataFix, along with the benchmarking methods used for feature shift localization and correction. The hyperparameter optimization involves conducting a grid search on the simulated datasets, with the provision that experiments exceeding a 30-hour runtime are excluded. For parameters left unspecified, we initialize them with their default values.

Table 4 outlines the search space for each parameter tuned in each detection benchmarking method, as well as their respective optimal value. Similarly, Table 5 provides insight into the search space for each parameter tuned in each correction benchmarking method, as well as the values that yielded optimal results.

DF-Locate and DF-Correct both undergo optimization with respect to the choice of classifier employed as a discriminator, as detailed in Section K. Additionally, we conduct a hyperparameter search for DF-Locate, focusing on variables such as window length and polyorder, which is further elaborated upon Section E.3.

Table 4: Search space and optimal values for tuned parameters in detection benchmarking methods.

| Method | Tuned Parameter | Search Space | Optimal |
|---|---|---|---|
| MI | threshold | [0.008, 0.01, 0.02, 0.05, 0.1] | 0.02 |
| selectKbest | significance_level | [0.008, 0.01, 0.02, 0.05, 0.1] | 0.01 |
| MB-SM, MB-KS | n_expectation | [30, 100, 500, 10000] | 30 |
| KNN-KS, Deep-SM | n_bootstrap_runs | [1, 5, 10, 25, 50, 250] | 250 |

## G.2  Evaluation Setting

In both shift localization and correction, each method only has access to the reference dataset $X$ and the corrupted query dataset $Y$ as input for the iterative process of training discriminators and computing heuristics. Note that the ground truth information, including the actual corrupted feature locations $C^*$ and the original (pre-shifted) query dataset $Y^*$, are not used by DataFix at any step when predicting the localization of corrupted features or when correcting the query dataset.

Table 5: Search space and optimal values for tuned parameters in correction benchmarking methods.

| Method | Tuned Parameter | Search Space | Optimal |
|---|---|---|---|
| KNN | k | [10, 25, 50, 100] | 10 |
| DD | n | [10, 50, 100, 200] | 100 |
| INB | n | [50, 100, 200, 500] | 200 |
| | ndim | [30, 60, 90] | 90 |
| MLP | activation | ['relu', 'tanh'] | 'relu' |
| | hidden_layer_sizes | [100, 1000] | 100 |
| | alpha | [0.0001, 0.001] | 0.0001 |
| | batch_size | ['auto', 64] | 'auto' |
| | learning_rate_init | [0.0001, 0.001] | 0.001 |
| GAIN | batch_size | [16, 64, 256] | 16 |
| | n_epochs | [1000, 50000] | 50000 |
| | hint_rate | [0.8, 0.9] | 0.9 |
| HyperImpute | baseline_imputer | [0, 1, 2] | 1 |
| | optimize_thresh | [1000, 5000] | 5000 |
| | n_inner_iter | [40, 80] | 40 |
| ICE | max_iter | [1000, 2000] | 1000 |
| | initial_strategy | [0, 1, 2] | 1 |
| MIRACLE | lr | [0.0001, 0.001] | 0.0001 |
| | batch_size | [1024, 512] | 1024 |
| | n_hidden | [16, 32, 64] | 16 |
| | reg_lambda | [0.1, 1, 10] | 10 |
| | reg_beta | [1, 3] | 3 |
| | window | [10, 20] | 20 |
| MissForest | n_estimators | [10, 20, 50] | 10 |
| | max_iter | [100, 500, 1000] | 100 |
| Sinkhorn | lr | [0.001, 0.01] | 0.01 |
| | n_epochs | [500, 1000] | 1000 |
| | batch_size | [256, 512] | 512 |
| | noise | [0.001, 0.01] | 0.001 |
| | scaling | [0.9] | 0.9 |
| SoftImpute | maxit | [1000, 2000] | 1000 |
| | convergence_threshold | [0.00001, 0.0001] | 0.0001 |
| | max_rank | [2, 3] | 3 |
| | shrink_lambda | [0.5, 0] | 0 |

# H Computational Time

Large and high-dimensional datasets are becoming the norm, therefore, methods that detect and correct feature shifts should be able to properly scale with respect to the number of samples and features.

Figure 13 presents the average computational time of feature shift localization benchmarking methods as a function of the product between the number of samples and features for each dataset. MI and selectKbest stand out as the fastest methods (using the Chi-square test for categorical datasets and ANOVA-F test for continuous datasets). MRMR and FAST-CMIM, although performing adequately in terms of speed for small datasets, encounter challenges in scaling with larger dataset sizes. Consequently, they fail to produce results within the 30-hour time limit for Founders and Canine datasets. Furthermore, the feature-shift detection techniques KNN-KS and, particularly MB-KS, demonstrate significantly slower performance, rendering them incapable of delivering results within the given time constraint for Founders, Canine, Dilbert, and Phenotypes datasets. DF-Locate proves to be a reasonably efficient method, exhibiting good scalability as the dataset size increases, while providing the best localization performance.

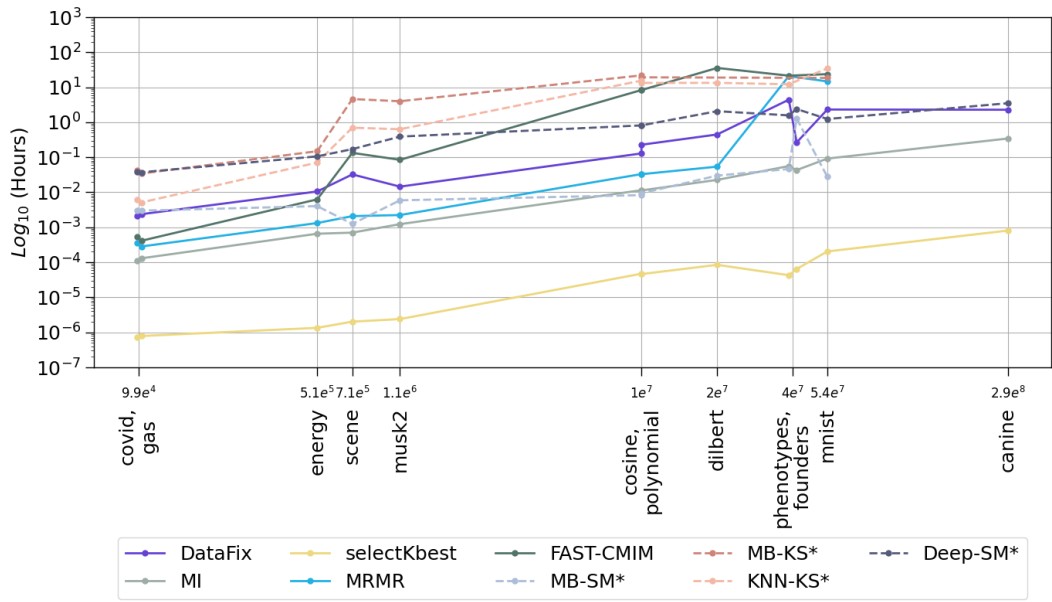

Figure 13: Computational time for shift localization methods based on real dataset size.

In Figure 14, we conduct a comparative analysis of the computational time for shift correction methods. Although DF-Correct is not faster than simpler techniques like median or linear regression, its speed surpasses several competing methods, such as MIRACLE and HyperImpute. Furthermore, DF-Correct exhibits reasonable runtime even for the largest high-dimensional Canine dataset, successfully correcting the distribution shift within the 30-hour time limit. Many of the competing methods were unable to provide results for the Canine dataset due to their excessive time complexity and/or memory requirements. Therefore, DF-Correct provides the best correction in terms of distribution shifts, while still providing competitive or faster speeds than competing methods.

All experiments were done with an Intel Xeon Gold with 12 CPU cores.

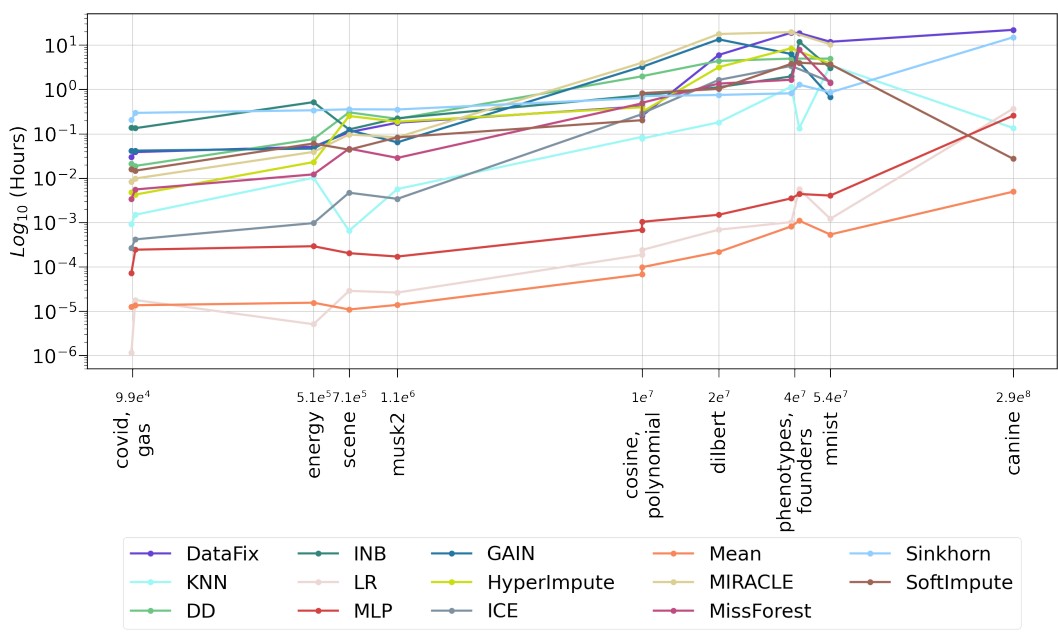

Figure 14: Computational time for shift correction methods based on real dataset size.

# I Extended Feature Shift Localization Results

We evaluate the performance of DF-Locate across different fractions of manipulated features: 5%, 10%, and 25%. The F-1 scores, depicted in Figure 15, were computed by averaging across manipulation types and taking the median across real datasets for each feature shift localization method. Our findings show that DF-Locate obtains consistently higher performance, irrespective of the fraction of manipulated features involved. However, it is important to note that this is not the case for other methods, such as MRMR and FAST-CMIM, which exhibit limitations in their localization capabilities, particularly when there are only a few corrupted features. Note that when a small percentage of manipulated features is present, a smaller amount of features need to be localized as corrupted. However, this could also lead, in some cases, to lower distribution shifts, making the detection of such shifts more challenging. On the other hand, the presence of a larger percentage of manipulated features can make the localization task more challenging, while the empirical detection of the presence of the shift can be, in some cases, easier.

Figure 16 shows the mean F-1 score of DF-Locate and various shift localization methods applied to the real datasets. The symbol 'x' denotes experiments that are missing due to exceeding the time limit of 30 hours. Note that most methods fail to process high-dimensional datasets such as Founders and Canine, whereas DF-Locate is able to provide accurate results while scaling to large datasets. DF-Locate consistently exhibits superior performance compared to all benchmarking methods across most datasets. The only exception observed is with the Phenotypes dataset, where selectKBest slightly outperforms DF-Locate in locating the corrupted features. Nevertheless, for the remaining datasets, DF-Locate surpasses all competing approaches and is only slightly surpassed or equaled in performance on a few occasions by Deep-SM (*) or KNN-KS (*), both of which use the ground truth $|C|$. Additionally, it is worth noting that MI always outperforms selectKbest on datasets with continuous features, while the opposite holds true for datasets with categorical features.

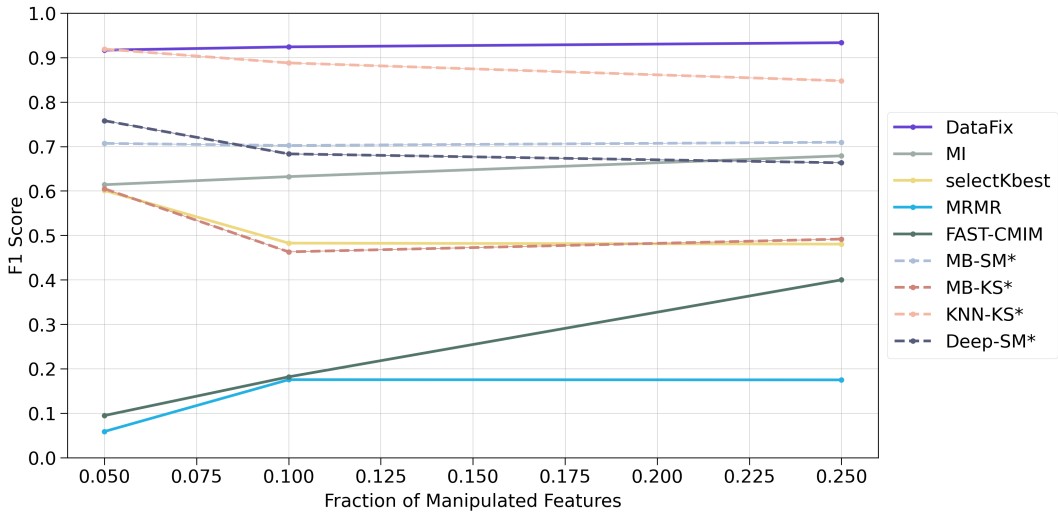

Figure 15: Median F-1 scores of shift localization methods by fraction of manipulated features on real datasets.

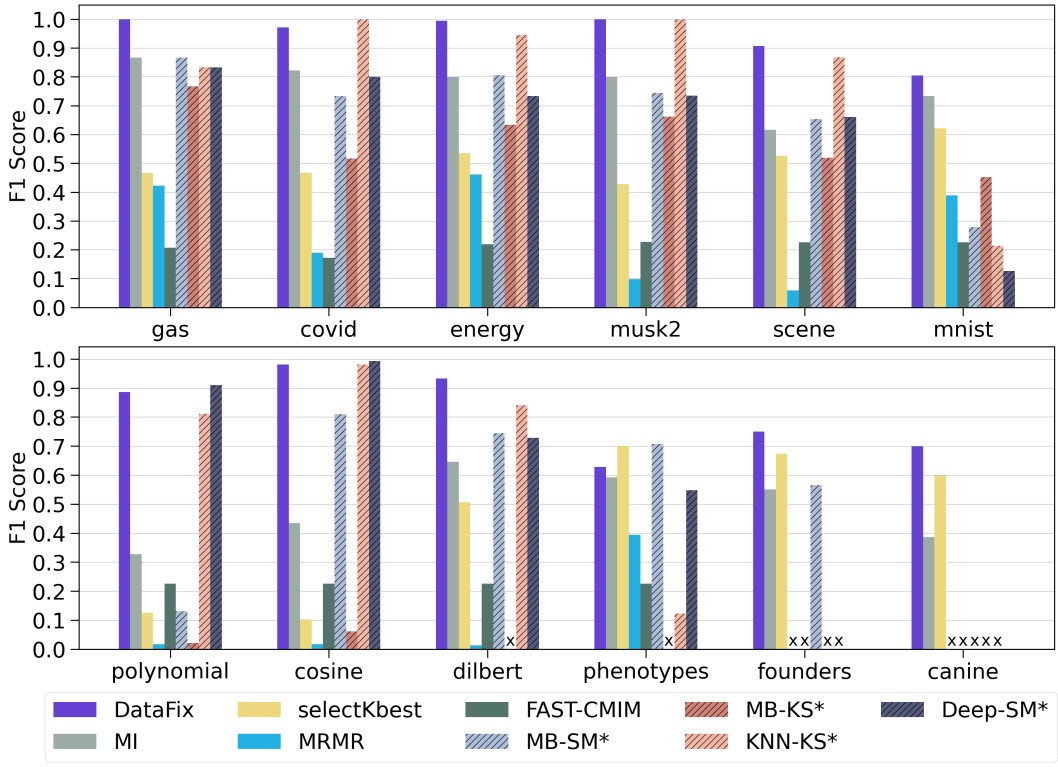

Figure 16: Mean F-1 scores of shift localization methods by real datasets. 'x' indicates missing experiment. Higher is better.

Figure 17 shows the median F-1 scores of DF-Locate and other competing methods, categorized by the feature manipulation type applied to the real datasets. The average F-1 score is computed across fractions of manipulated features, followed by the computation of the median F-1 score across different datasets. The symbol 'x' is used to indicate missing experiments for MB-KS and manipulation types 6.1-6.3 and 10. These manipulations are applied to categorical datasets only (Phenotypes, Founders, Canine), and the MB-KS method did not yield results for any of these three datasets within the specified time constraint of 30 hours. Manipulations involving shifts caused

by the correlation between features (manipulations 3 and 8) are not detected by methods such as MI, selectKbest, MRMR and Fast-CMIM, while being accurately detected by MB-SM, KNN-KS, Deep-SM, and our proposed method. Note that while manipulation type 8 breaks the theoretical equivalence between feature selection and feature shift localization problem (see previous sections), it is still accurately localized by DataFix. Manipulation 10, consisting of replacing feature values with the ones predicted with a KNN, is the most challenging to detect by DataFix. Note that such manipulation can lead, in some scenarios, to small or undetectable distribution shifts, as KNN provides perfect predictions when its training dataset size goes to infinity.

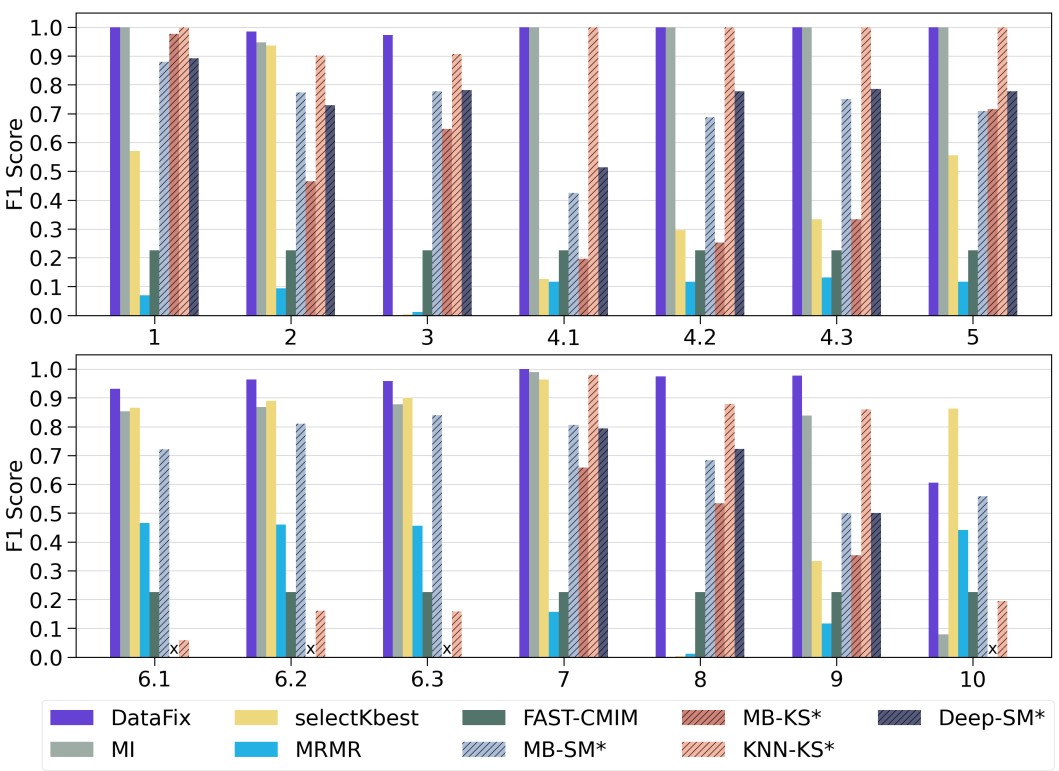

Figure 17: Median F-1 scores of shift localization methods by feature manipulation type on real datasets. 'x' indicates missing experiment. Higher is better.

Figure 18 displays the mean F-1 scores of shift localization methods across the simulated datasets. As observed in the results presented for real datasets in Figure 16, DF-Locate consistently outperforms or matches all competing methods, except for datasets 7 and 8, where MB-SM and Deep-SM obtain a higher F-1 localization score when using $|C|$ as extra ground truth information. Note that in a fair comparison where $|C|$ is not used, MB-SM and Deep-SM perform poorly (see Figure 3). DF-Locate obtains lower F-1 scores in simulated datasets 5, 7, and 8 compared to the other datasets, which involve shifts originated by a mismatching correlation between distributions.

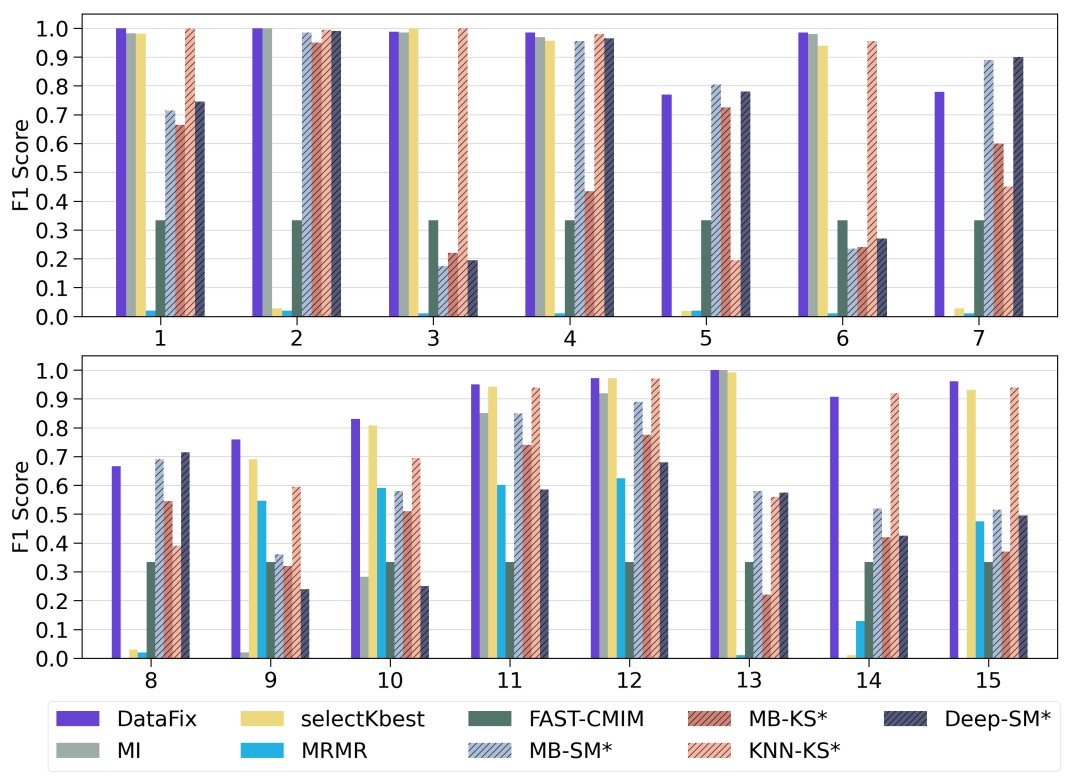

Figure 18: Mean F-1 scores of shift localization methods on simulated datasets. Higher is better.

## J    Extended Feature Shift Correction Results

Figures 19 and 20 provide a comprehensive evaluation of the performance of DF-Correct and competing shift correction methods across real datasets, by using the Wasserstein distance ($W_2^2$) as in [48], the empirical estimator of the Henze-Penrose divergence ($D_{hp}$) [85, 86, 87], and the empirical estimator of the symmetric Kullback-Leiber divergence ($D_{skl}$) [88]. $W_2^2$, $D_{hp}$, and $D_{skl}$ are non-parametric estimators of distances between the underlying distributions $p$ and $q$ that make use of the reference $X$ and query $Y$ datasets. By computing statistics based on distances between samples within and between datasets, these metrics estimate the distance between distributions. The reported metrics provide empirical estimates of the divergences between the corrected query datasets and the reference datasets. Note that first, the background empirical divergences between the reference and query dataset (prior to any manipulation) are computed and subtracted from the reported metrics. For simulated datasets where there are no query datasets prior to manipulation, a second reference dataset is used to obtain background empirical divergences.

As shown in Figures 19 and 20, DataFix outperforms all other methods by a significant margin for most datasets, demonstrating its high effectiveness in providing corrected query datasets that are close to the reference distribution. Following DataFix, simpler methods like KNN and linear regression demonstrate competitive performance for most datasets, while techniques such as ICE, HyperImpute, INB, Sinkhorn, and MLP yield favorable results for specific datasets. Furthermore, DataFix consistently achieves the lowest $W_2^2$ values across most datasets, with exceptions being MNIST and Phenotypes. Specifically, for the MNIST dataset, KNN, MLP, GAIN, HyperImpute, and ICE surpass DataFix achieving the best $W_2^2$ value of 0. Note that for high-dimensional datasets such as Dilbert, Phenotypes, Founders, and Canine, the $W_2^2$ metric saturates to 0 for multiple methods, making the other metrics a better alternative to compare the quality between methods.

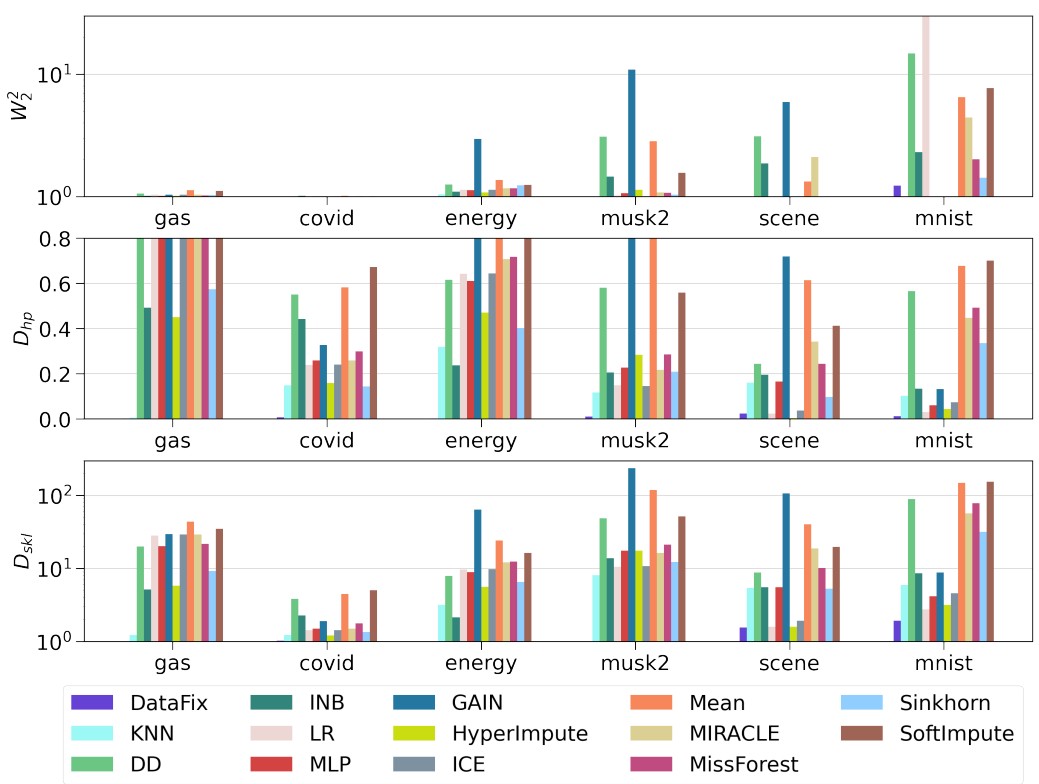

Figure 19: $W_2^2$, $D_{hp}$, and $D_{skl}$ of shift correction methods on real datasets. Lower is better. (Part 1)

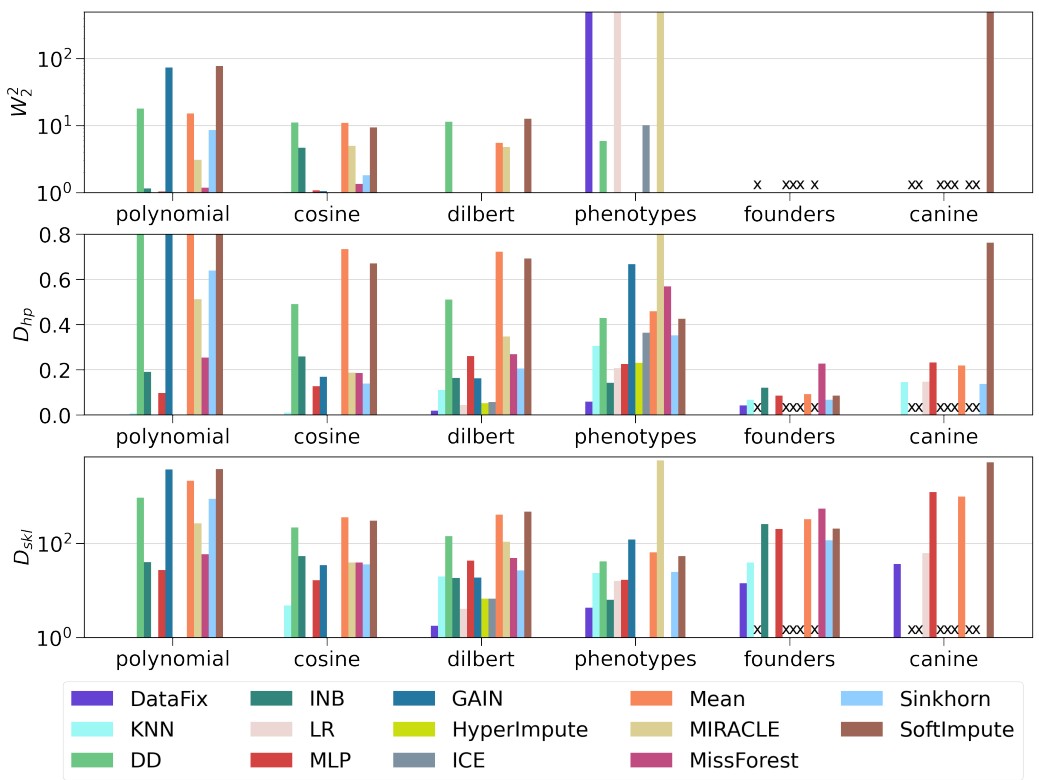

Figure 20: $W_2^2$, $D_{hp}$, and $D_{skl}$ of shift correction methods on real datasets. Lower is better. (Part 2)

# K  Classifier Analysis for Localization and Correction

Figure 21 provides the F-1 score results for DF-Locate when using different classifiers as discriminators within the iterative process on both real and simulated datasets. Tree-based methods including Random Forest (RF), CatBoost, ExtraTree, and LightGBM (LGBM) provide highly similar results, with high F-1 scores, surpassing linear models such as logistic regression (LogReg) and a support vector classifier (SVC). We selected RF as our discriminator as it provided much faster training times while being highly competitive in localization accuracy.

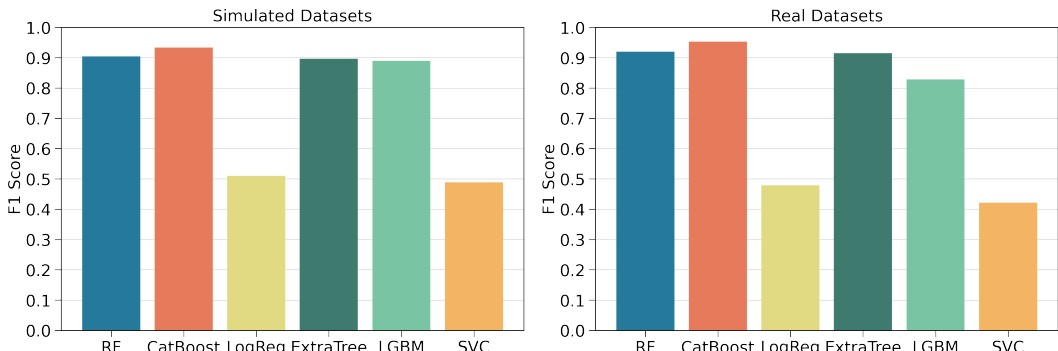

Figure 21: Mean F-1 scores by DF-Locate using different classifiers on simulated (left) and real (right) datasets. Higher is better.

Figure 22 provides the feature shift correction metrics for DF-Correct when using different classifiers as discriminators within the iterative process on both real and simulated datasets. Similar to DF-Locate, tree-based methods provide similar results, surpassing the linear methods in most metrics. We use $D_{hp}$ as a metric to select our method because it provides an estimate that tightly bounds the total variation distance. Namely, we select CatBoost as our discriminator as it provides competitive performance with the other tree-based methods in the simulated datasets, and clearly outperforms the others in the real datasets.

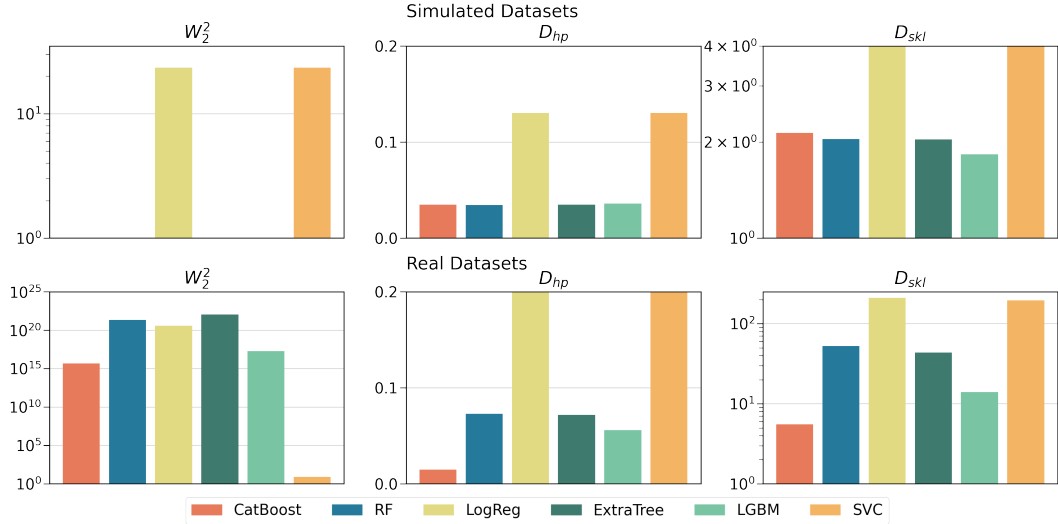

Figure 22: Mean $W_2^2$, $D_{hp}$, and $D_{skl}$ by DF-Correct using different classifiers on simulated (top) and real (bottom) datasets. Lower is better.

# L  Variability Analysis for Location and Correction

Conducting the whole evaluation benchmark for both localization and correction multiple times is infeasible and expensive. However, to provide insight into the variation among runs for each method, we present plots showcasing the variability of detection and correction methods with multiple random seeds for a given dataset, namely the Energy dataset.

In Figure 23, we present the variability of detection methods, conveyed by the mean and standard deviation of the F-1 score across five different seeds on the Energy dataset. This evaluation covers scenarios involving 25% corrupted features and all manipulation types. Similarly, Figure 24 provides insights on the variability of correction methods, conveyed by the mean and standard deviation of $W_2^2$, $D_{hp}$, and $D_{skl}$ computed across five different seeds on the Energy dataset. All detection and correction methods, particularly DF-Locate and DF-Correct, exhibit a low variability across different seeds.

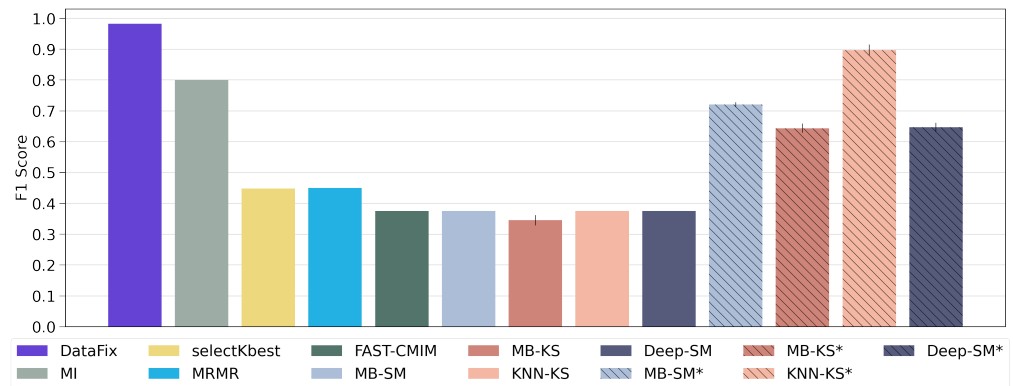

Figure 23: Variability of detection methods on the Energy dataset. Mean and standard deviation of F-1 score across five different seeds, for 25% corrupted features and all manipulation types. Higher is better.

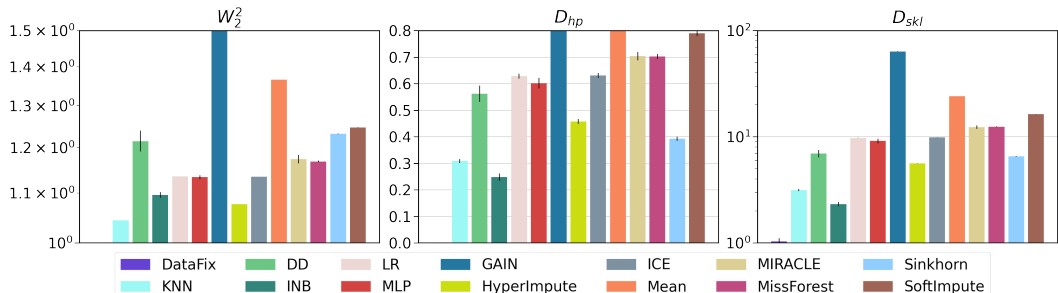

Figure 24: Variability of correction methods on the Energy dataset. Mean and standard deviation of $W_2^2$, $D_{hp}$, and $D_{skl}$ across five different seeds. Lower is better.

# M  Evaluation of Downstream Classification and Regression Tasks after Correction

In this section, we include an evaluation of downstream classification and regression tasks for the Musk2 and Energy datasets, which include categorical and continuous labels, respectively.

We train the downstream models with the corrected query datasets and evaluate their performance on the reference dataset. We perform the experiment using each query dataset version, which includes the original pre-corrupted dataset, and the correct datasets with each correction method. The evaluation of the classifier/regressor is carried out under three different scenarios: when all features are used (100%), when only 50% of the non-corrupted features are included, and when 0% of non-corrupted features are included (only the corrected features are used). The classification results are reported in Table 6, and the regression results are reported in Table 7, using balanced accuracy for classification and root mean square error (RMSE) for regression. Methods are sorted by their performance without non-corrupted features (0%). Notably, the classification/regression downstream performance when using DataFix surpasses the competing methods in most settings, in some cases even surpassing the performance of directly using the original query dataset (ground truth pre-corruption). This observation demonstrates that the class-conditional divergence between the reference and the corrected query with DataFix is small.

Table 6: Downstream classification task with Musk2 dataset, with all corrected features and all non-corrupted features (100%), with half of the non-corrupted features (50%), and without non-corrupted features (0%).

| Method | Balanced Acc. (0%) | Balanced Acc. (50%) | Balanced Acc. (100%) |
|---|---|---|---|
| Original Query | **0.951** | **0.960** | **0.964** |
| DataFix | 0.902 | 0.953 | 0.957 |
| LR | 0.871 | 0.932 | 0.938 |
| HyperImpute | 0.863 | 0.932 | 0.945 |
| ICE | 0.848 | 0.937 | 0.937 |
| MIRACLE | 0.828 | 0.920 | 0.937 |
| Sinkhorn | 0.818 | 0.924 | 0.936 |
| MLP | 0.817 | 0.907 | 0.911 |
| KNN | 0.815 | 0.925 | 0.934 |
| INB | 0.812 | 0.950 | 0.954 |
| MissForest | 0.779 | 0.908 | 0.944 |
| DD | 0.576 | 0.950 | 0.956 |
| GAIN | 0.574 | 0.910 | 0.939 |
| Mean | 0.500 | 0.952 | 0.961 |
| SoftImpute | 0.493 | 0.620 | 0.836 |

Table 7: Downstream regression task with Energy dataset, with all corrected features and all non-corrupted features (100%), with half of the non-corrupted features (50%), and without non-corrupted features (0%).

| Method | RMSE (0%) | RMSE (50%) | RMSE (100%) |
|---|---|---|---|
| DataFix | **0.089** | **0.082** | **0.080** |
| Original Query | 0.093 | 0.084 | 0.082 |
| KNN | 0.105 | 0.094 | 0.091 |
| Sinkhorn | 0.107 | 0.090 | 0.085 |
| INB | 0.114 | 0.094 | 0.092 |
| Mean | 0.122 | 0.088 | 0.083 |
| GAIN | 0.127 | 0.200 | 0.279 |
| HyperImpute | 0.129 | 0.124 | 0.123 |
| DD | 0.137 | 0.120 | 0.116 |
| MissForest | 0.147 | 0.155 | 0.156 |
| MLP | 0.175 | 0.187 | 0.200 |
| SoftImpute | 0.180 | 0.182 | 0.252 |
| LR | 0.191 | 0.189 | 0.205 |
| ICE | 0.217 | 0.225 | 0.234 |
| MIRACLE | 0.311 | 0.335 | 0.343 |

## N    End-to-end DataFix Analysis

Figure 25 provides five additional examples that illustrate the iterative process of DF-Locate before and after shift correction, in simulated datasets 1, 2, 3, 6, and 10. In each dataset, there are 200 corrupted features out of 1000. Similar to Figure 5, the left column in Figure 25 displays the TVD estimated by the random forest (blue), which provides a lower bound for its ground truth Monte Carlo estimate (black), as the iterative process detects and removes corrupted features. The F-1 detection score progressively increases until all corrupted features are identified, leading to the termination of the iterative process. The right column in Figure 25 showcases the iterative process applied to the corrected query using different methods.

In simulated datasets 1 and 2, the methods DD, INB, MIRACLE and MLP all yield an updated query that results in a lower empirical divergence. Notably, DD stands out by providing a corrected query with no empirical divergence detected by DF-Locate. However, DD produces queries with considerably higher empirical divergence in the case of the other simulated datasets when compared to DF-Correct, and is not able to lower the empirical divergence for simulated dataset 10. Moving to simulated dataset 6, INB and DD yield an updated query that reduces the empirical divergence, with both techniques achieving an empirical divergence that is almost negligible. The other shift correction benchmarking methods generate an updated query that increases the shift instead of reducing it. In simulated dataset 10, none of the benchmarking methods are able to generate an updated query that leads to a lower empirical divergence. Remarkably, DF-Correct (Purple) provides a precisely corrected query with no empirical divergence detected by DF-Locate for all datasets.

## O    DataFix Limitations

DataFix is specifically tailored for tabular datasets and will not work optimally when applied to other data types such as images, videos, audio, speech, or textual data. DataFix also exhibits limitations due to its computational cost, preventing scalability for online or streaming scenarios, and potentially resulting in reduced speed with very large datasets, although it generally outperforms competing methods in terms of computational efficiency.

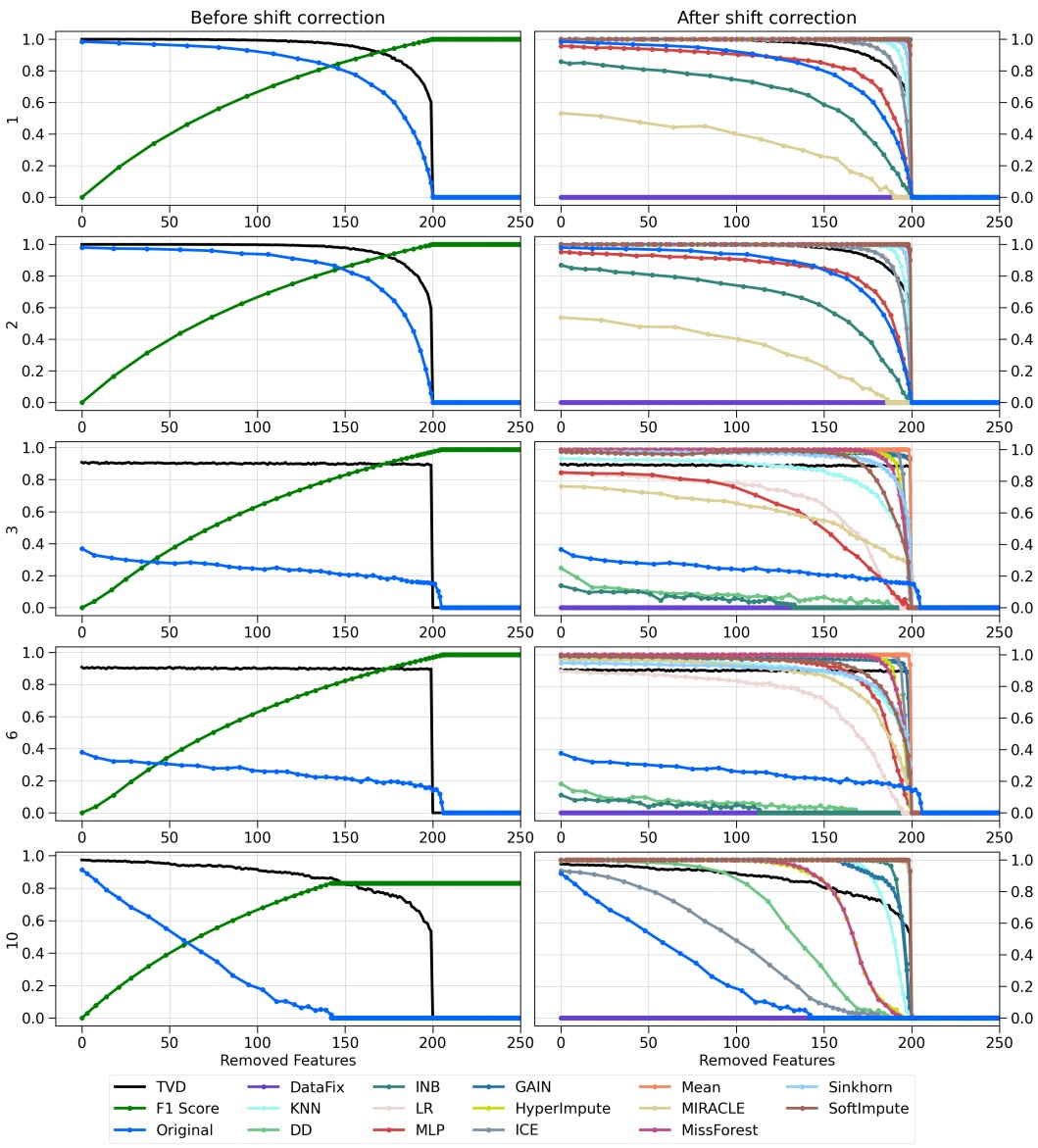

Figure 25: DF-Locate iterative process before (left) and after (right) shift correction. Each row corresponds to a different simulated dataset.

