# OpenReview forum: "Adversarial Learning for Feature Shift Detection and Correction"
_NeurIPS.cc/2023/Conference — NeurIPS 2023 poster_

### Official Review · Reviewer_5LeU · 2023-06-13

**Soundness:** 2 fair
**Presentation:** 3 good
**Contribution:** 2 fair
**Rating:** 5
**Confidence:** 3

**Summary:**

This paper introduces an invariance-based approach to out-of-distribution generalization where, for a given pair of distributions or domains $p$ a $q$, a subset of the input dimensions is filtered away so as to minimize some divergence between the filtered data, and then classifiers trained on samples from $p$ generalize to samples from $q$. The proposal leverages feature selection approaches to find a subset of the input space dimensions that minimizes a divergence between modified $p$ and $q$, as estimated by a domain discriminator. In addition, an adversarial approach is further introduced where features deemed responsible for the shift are corrected.

**Strengths:**

- A very relevant problem is tackled with potential to practical relevance.
- The manuscript is very clearly written and easy to follow.
- The proposal applies well established approaches to features selection and estimation of distribution shifts.

**Weaknesses:**

- Lack of contextualization with out-out-distribution generalization literature. Is the setting under consideration discussed in the OOD generalization literature? If so, it should be highlighted. It seems the type of shift under consideration is the standard covariate shift case, where data marginal distributions shift (in a particular way since only a subset of the input dimension shifts), and the divergence used for feature selecting is the standard H-divergence estimator (prediction accuracy of a domain discriminator) [1].
- Potentially hidden assumptions: extra assumptions over data conditional label distributions are required since resulting pruned features are domain invariant. In other words, $p(y|x \sim p) = p(y|x \sim q)$, which is not stated. Refer to [1,2] for discussion on settings and corresponding assumptions for OOD generalization cases.
- Empirical assessment limited to tabular data, and limited to a particular pair of distributions. That is, the resulting set of features is only invariant across the pair of domains used for selection, and little is known about what would happen had a new shifted distributed been observed.
- The proposal and the evaluation ignore representation learning techniques, which are an alternative to feature selection. Invariance-based approaches to out-of-distribution robustness typically project data onto a space where domains cannot be discriminated, with domain-adversarial approaches being a popular example of such an approach [3, 4].
- The motivation to correction approach is unclear to me. Are the resulting data any useful/meaningful after the correction approach is performed?

[1] Ben-David, Shai, John Blitzer, Koby Crammer, Alex Kulesza, Fernando Pereira, and Jennifer Wortman Vaughan. "A theory of learning from different domains." Machine learning 79 (2010): 151-175.

[2] Zhao, Han, Remi Tachet Des Combes, Kun Zhang, and Geoffrey Gordon. "On learning invariant representations for domain adaptation." In International conference on machine learning, pp. 7523-7532. PMLR, 2019.

[3] Ajakan, Hana, Pascal Germain, Hugo Larochelle, François Laviolette, and Mario Marchand. "Domain-adversarial neural networks." arXiv preprint arXiv:1412.4446 (2014).

[4] Albuquerque, Isabela, João Monteiro, Mohammad Darvishi, Tiago H. Falk, and Ioannis Mitliagkas. "Generalizing to unseen domains via distribution matching." arXiv preprint arXiv:1911.00804 (2019).

**Questions:**

- How does the proposal assume labels shift across distributions/domains?
- How does features selection compare with invariant representation learning?
- Does inducing invariance across a pair of domains $p$ and $q$ influence invariance across other domains? Under which conditions?
- How would the feature selection approach fare in more complex structured data, such as images or text for instance? On could first encode the data with a pre-trained model and apply the proposal on top of resulting features.
- Could the authors clarify a bit the motivation on the correction approach? One would modify their data so it's "less different" than what is observed from a different data source, but then are the resulting corrected data more useful than the invariant subset of the features?

**Limitations:**

Cf. weaknesses section for details. Main limitations are lack of contextualization with previous work on OOD generalization, and a limited evaluation. It's also unclear what would be the motivation for a correction approach such as the one proposed.

---

> ### Author Rebuttal · Authors · 2023-08-09
>
> **Dear reviewer,**
>
> **Thank you for your constructive suggestions. We believe there has been misunderstanding regarding the goals and scope of the paper. We introduce two systems: the first localizes faulty features, which can be used to detect incorrectly standardized or processed values or to localize malfunctioning sensors in multi-sensor environments. The second system proposes new values for the corrupted data, by performing divergence-reducing “imputation” of the data. All classifiers described in the paper are discriminators trained to classify between real and corrupted as part of the iterative processes of localization and correction.**
>
> **Thus, your summary line “... and then classifiers trained on samples from p generalize to samples from q” is not performed at any point throughout the paper. While the proposed method could be used as a way to provide more homogenized datasets to train ML systems, training classifiers with out-of-distribution generalization is out of the scope of this paper and not directly comparable to our approaches. In fact, many of the datasets used do not have semantic labels, so training a classifier/regressor with them would not in any case be possible. Feature shift localization (and correction) is useful in many applications and settings and is an independent problem from OOD generalization. The paper [NeurIPS 2020] provides a similar motivation to our one, and we use it as our benchmark baseline.**
>
> &nbsp;
>
> *“... Is the setting under consideration discussed in the OOD generalization literature? If so, it should be highlighted ...”*
>
> **There has been a misunderstanding with respect to the problem that we are solving; OOD generalization is not directly comparable to our work. Our setting provides a mechanism to “detect” data errors and “fix” them, in an application-independent manner.**
>
> **As feature shift localization (and correction) is related to multiple fields of statistics and ML, we are including an extended related work section in the appendix, where we are mentioning OOD generalization, outlier detection, and GANs among others.**
>
> &nbsp;
>
> *“... extra assumptions over data conditional label distributions are required since resulting pruned features are domain invariant. In other words, p(y|x∼p)=p(y|x∼q), which is not stated.”*
>
> **Our setting does not include any classification tasks, and there is no training of supervised ML methods for prediction. The “y” (label) is not part of our framework, in fact, most of the datasets do not have semantic labels.**
>
> &nbsp;
>
> *“Empirical assessment limited to tabular data, and limited to a particular pair of distributions.”*
>
> **Our work focused on tabular data as it is where the problem of feature shift localization most commonly arises. For example, this system is being used to homogenize tabular datasets for medical statistical analysis. We adopt a similar framework as in [NeurIPS 2020] where only two distributions are considered. The proposed method can be applied to more distributions/data sources by applying the system iteratively.**
>
> &nbsp;
>
> *“The proposal and the evaluation ignore representation learning techniques... Invariance-based approaches to out-of-distribution robustness typically project data onto a space where domains cannot be discriminated ...”*
>
> **We do not consider out-of-distribution robustness / representation learning as they are not directly comparable. Those techniques learn a new space that provides better generalization across domains for classification. Here we try to localize and fix erroneous features within datasets.**
>
> &nbsp;
>
> *“Are the resulting data any useful/meaningful after the correction approach is performed?”*
>
> **The correction approach can be seen as a “divergence-removing” imputation. The usefulness and meaningfulness of the corrected dataset will vary largely depending on the application and type of data, in a similar way to data imputation.**
>
> &nbsp;
>
> *“How does the proposal assume labels shift across distributions/domains?”*
>
> **We don’t solve a classification problem and there are no labels.**
>
> &nbsp;
>
> *“How does features selection compare with invariant representation learning?”*
>
> **While our system can generate datasets that could provide more invariance for downstream classification, our approach differs from invariant rep. Learning, as this learns a representation of the input so it has invariance between domains. Here we try to localize and correct a dataset with corrupted features.**
>
> &nbsp;
>
> *“Does inducing invariance across a pair of domains p and q influence invariance across other domains? Under which conditions?”*
>
> **We are trying to detect faulty features and provide corrected datasets; we are not learning an invariant representation.**
>
> &nbsp;
>
> *“How would the feature selection approach fare in more complex structured data, such as images or text for instance?”*
>
> **Applying feature selection methods directly to natural images could lead to poor results. However, the problem we are trying to solve arises most commonly in tabular data.**
>
> &nbsp;
>
> *“Could the authors clarify a bit the motivation on the correction approach? One would modify their data so it's "less different" than what is observed from a different data source, but then are the resulting corrected data more useful than the invariant subset of the features?”*
>
> **Similar to data imputation, the usefulness of the corrected features will depend on the nature of the data and the application at hand. For example, in order to compute statistics and correlations between features, using a corrected dataset can help to reduce some bias that might appear if the corrupted dataset is used.**
>
> &nbsp;
>
> **We hope that we have been able to clarify the goal and scope of the paper and you will consider increasing the score.**
>
>
> **[NeurIPS 2020] Kulinski, Sean, Saurabh Bagchi, and David I. Inouye. "Feature shift detection: Localizing which features have shifted via conditional distribution tests."**

---

> > ### Comment · Reviewer_5LeU · 2023-08-10
> > **Response to authors**
> >
> > Thank you for the clarification. Indeed, there has been misunderstanding on my end, likely due to similarity between the proposal and invariant representation learning approaches. In particular, the reference/query classifier estimates what that robustness literature often refers to as the H-divergence, and learning representations minimizing that divergence is a common approach to enable training predictors on a data source and testing on another. I clarify that dropping shifting-causing features is what I refer to as invariant representation learning, but realize now that's not in scope, and only localization is tackled (plus fixing the shifting dimensions). I would still recommend for authors to contextualize their work with respect to that literature given the similarity. I do now have concerns regarding the scope being too narrow, though considering the clarifications and the other reviews, I will increase my score.

---

> > > ### Author Response · Authors · 2023-08-10
> > >
> > > **Thank you for reading our rebuttal, providing such a fast response, and increasing the score!**
> > >
> > > **We completely agree that feature shift localization/correction has some similarities with invariant representation learning, and we are including a detailed extended related work section in the appendix describing them, as well as providing some related literature that was left out due to space limitations in the main text. We hope that this can inspire future work looking deeper into the relationship of feature selection, shift localization, and invariant representation learning. Regarding the scope of the paper, we adopted a similar scope and framework as in [NeurIPS 2020], which we believed was a good approach for this work.**
> > >
> > > **[NeurIPS 2020] Kulinski, Sean, Saurabh Bagchi, and David I. Inouye. "Feature shift detection: Localizing which features have shifted via conditional distribution tests."**

---

### Official Review · Reviewer_MQtc · 2023-06-19

**Soundness:** 2 fair
**Presentation:** 2 fair
**Contribution:** 3 good
**Rating:** 6
**Confidence:** 2

**Summary:**

The authors tackle the problems of feature shift localization and correction, i.e., identifying columns leading to divergence between two distributions, and imputing new values in their place, leading to lower divergence. For the first task they employ a random forest classifier trained to predict between samples coming from the reference or query distributions. The corrupted features are identified based on the feature importance scores of the random forest models.

For the correction task, a set of proposed corrections is generated per-sample, and the correction yielding the lowest probability under a classifier trained to distinguish between the distributions is selected as the imputation.

**Strengths:**

1. The proposed method is outperforming all the baselines
2. the underlying models are relatively simple and computationally efficient
3. The authors provided the full code in the supplementary

**Weaknesses:**

1. Most importantly, how were train/test splits performed? The text implies performance is measured directly on the train tests. Analyzing the code for the feature correction task it seems that the models were indeed evaluated directly on the training data.
2. Several experimental details are missing. Why was the optimal hyperparameter configuration selected only for a subset of the baselines in the feature localization tasks? How were hyperparameters selected for the baselines for the feature correction tasks?
3. The paper could be written in a clearer way at times. Especially the description of both methods (Sections 4 and 5) are somewhat convoluted, in particular the last paragraph of Section 5. what are the **k** classifiers and the CatBoost classifiers trained to predict? How are the values of B generated? How is the objective of equation 9 optimized? The method could have been described clearer in the main text, instead of describing e.g., the total variance distance for the case not applicable to practice.
4. Error bars of the results are not present, contrary to what has been specified by authors

Please note that my rating of the paper is mostly influenced by points 1 and 2, as they seem to invalidate the empirical results presented by authors.

**Questions:**

The main weakness of the paper are the missing experimental details, i.e., points 1. and 2. of Weaknesses.

For 1., it could very well be that I misunderstood the code and/or the text. If so, please indicate the fragments of attached codebase, were the train/test splits are indeed performed. In my understanding, judging for example by lines 461-470 in correct/src/scripts/run_benchmarks.py, the same dataset is used for model fitting as well as evaluation.
For 2., authors should clarifiy how they selected hyperparameters for all baseline methods.

Other questions:

1. What is the motivation behind definining the threshold as $\kappa D$ (lines 213-214 in text)? Is it not possible for a single feature to cause high divergence?

**Limitations:**

-

---

> ### Author Rebuttal · Authors · 2023-08-08
>
> **We want to thank the reviewer for the insightful comments. We want to address each of the questions and concerns:**
>
> *“Most importantly, how were train/test splits performed? The text implies performance is measured directly on the train tests. Analyzing the code for the feature correction task it seems that the models were indeed evaluated directly on the training data.”*
>
> **We believe that the reviewer has a perfectly reasonable misunderstanding. In the task of feature shift correction, there is no need for a train/test split. The evaluation process is similar to data imputation evaluation: algorithms take as input a matrix (concatenated reference+query) with some missing (or corrupted) values, and the algorithm outputs the same matrix with the missing/corrupted values replaced. The evaluation is done by taking the imputed/corrected matrix (the output matrix) and comparing it with the matrix before any corruption/missingness was present (the ground truth matrix). As long as the algorithm does not see the “ground truth matrix” during the prediction of the “output matrix”, and only sees the “corrupted input matrix”, the evaluation setting is perfectly valid, and it is the common setting adopted in missing data imputation tasks. Both shift detection and correction tasks follow a similar setting, where the training/testing split is not required. As our algorithms train classifiers internally, as part of the filtering and correction process, we acknowledge that there can be confusion regarding the lack of common train/test split used in regular supervised classification/regression tasks, therefore, we will add some clarifying text in both the main text and appendix to avoid causing any confusion to the readers.**
>
> &nbsp;
>
> *“Several experimental details are missing. Why was the optimal hyperparameter configuration selected only for a subset of the baselines in the feature localization tasks? How were hyperparameters selected for the baselines for the feature correction tasks?”*
>
> **We selected the optimal hyperparameters by doing a grid search for all methods in both localization and correction tasks. We acknowledge that this was not properly explained in the paper so we are including this information in the main text, and all the details of the search in the appendix. Note that in the localization task, some methods are marked with “*” indicating that the methods make use of the number of corrupted features as extra input, making the comparison between the other methods not fair.**
>
> &nbsp;
>
> *“The paper could be written in a clearer way at times. Especially the description of both methods (Sections 4 and 5) are somewhat convoluted, in particular the last paragraph of Section 5. The method could have been described clearer in the main text, instead of describing e.g., the total variance distance for the case not applicable to practice.”*
>
> **Thanks for the comment! We agree that a lot of the theoretical part could be moved in the appendix and more details could be included in the main text. We are restructuring the paper to include a more detailed description of the methods in the main text.**
>
> &nbsp;
>
>  *“what are the k classifiers and the CatBoost classifiers trained to predict?”*
>
> **All the classifiers trained in localization and correction tasks are trained to classify the samples as “corrupted” vs “non-corrupted” (i.e. reference vs query).**
>
> &nbsp;
>
> *“How are the values of B generated?”*
>
> **B is a set that consists of the feature values within the reference, the feature values after performing imputation with linear regression and nearest neighbor, and a shuffle of feature values from the reference panel. We are adding an additional description of its construction in the appendix.**
>
> &nbsp;
>
> *“How is the objective of equation 9 optimized?”*
>
> **Equation 9 is a simple combinatorial problem. We use the discriminator to evaluate every sample with each proposal b in B, and simply select the b that provides the highest probability of being non-corrupted.**
>
> &nbsp;
>
> *“Error bars of the results are not present, contrary to what has been specified by authors”*
>
> **Thank you for pointing this out; we did not provide a plot with error bars. While performing the whole evaluation benchmark for both localization and correction multiple times is infeasible and expensive (as it would require around 2 months of computing resources), we are including a plot when performing the evaluation of both localization and detection with multiple random seeds for a given dataset, showcasing the variability between runs for each method. The plot is added in the appendix and provided in the attached pdf (attached figures 3 and 4).**
>
> &nbsp;
>
> *“What is the motivation behind definining the threshold as kD (lines 213-214 in text)? Is it not possible for a single feature to cause high divergence? “*
>
> **After normalizing and sorting the features, we select the features with the highest importance until their cumulative sum surpasses tD. Where 0<“t”<1 is a fixed hyperparameter and “D” is the empirical divergence (which is typically non-decreasing). You can imagine “D” as acting as a “learning rate scheduler” where, as the filtering process evolves, forces that fewer features are filtered at each step, to avoid incorrectly removing incorrect features, and “t” acts as a scaling factor. It is completely possible to have a single feature causing high divergence. If that is the case, it is likely that the normalized sorted features will look like a 1 followed by 0s (Note that the normalized and sorted features add to 1), because tD < 1, the first feature will be selected but not the following ones, leading to a correct detection.**
>
> &nbsp;
>
> **We hope that we have addressed all your doubts and concerns, that we have provided enough arguments showing the validity of the adopted evaluation setting, and that you will consider increasing the score.**

---

> > ### Comment · Reviewer_MQtc · 2023-08-10
> >
> > Thank you for your clarifications.
> > My two main concerns were regarding:
> > 1. The lack of train/test splits.
> > 2. Experimental details.
> >
> > As for 1., as I am not deeply familiar with the literature on data imputation, I will trust the authors that it is a common (and valid) evaluation practice.
> >
> > As for 2., you mention you performed a grid search over all hyperparameter settings. Based on which metric did you select the optimal settings?

---

> > > ### Author Response · Authors · 2023-08-10
> > >
> > > **Thank you for reading our reply and providing such a fast response!**
> > >
> > > *Based on which metric did you select the optimal settings?*
> > >
> > > **We used the F-1 score to select the best hyperparameters for the feature shift localization task. For the feature shift correction task, we used the HP-Divergence (D_hp). We use this one instead of W2 and Symmetric KL Div for two reasons: (1) HP-divergence has a bounded range (goes from 0 to 1). On the other hand, W2 and Symmetric KL values can be very large for some methods and datasets, and lead to biased results (averaged across datasets) for some baselines.  (2) the HP-Divergence can be used to provide lower and upper bounds of the total variation divergence (which is the one used throughout the theoretical setting). We have included all these details in the main text, as well as a better justification for the selected evaluation metrics in the appendix.**
> > >
> > > **We hope that this addresses all your concerns, and you will consider upgrading the score. Please let us know if you have any further questions or suggestions.**

---

> > > > ### Comment · Reviewer_MQtc · 2023-08-11
> > > >
> > > > Thank you for your response.
> > > >
> > > > I assume the F1 score and the HP divergence were computed using the ground-truth matrix, correct?

---

> > > > > ### Author Response · Authors · 2023-08-11
> > > > >
> > > > > **Thank you for your reply!**
> > > > >
> > > > > **Yes, that is correct! The F-1 (in the localization task) was computed by comparing the ground-truth positions of manipulated features, and the predicted positions. The HP-Divergence (in the correction task) was computed using the corrected query matrix and the ground truth (i.e., pre-manipulation) matrix.**

---

> > > > > > ### Comment · Reviewer_MQtc · 2023-08-11
> > > > > >
> > > > > > Thanks for your answer.
> > > > > > In that case however I still find the experimental protocol to be incorrect. Since you select the hyperparameters based on performance on all datasets, there is are no guarantees as to the generalization abilities of your algorithm. This would be different if you were to hold out a set of datasets, or actually implement train/test splits and find the hyperparameters on the training sets

---

> > > > > > > ### Author Response · Authors · 2023-08-11
> > > > > > >
> > > > > > > Thanks for the response! **To be clear we do not select the hyperparameters based on performance on all datasets; we do use hold-out datasets**. We had thought we provided this information in your individual rebuttal response, but we now see we included this information only in the **global author rebuttal response** above, apologies. **We use the simulated datasets (probabilistic datasets) to do the hyperparameter search, and the real datasets are used purely for testing.** We already updated the manuscript to be particularly clear on this point, so as to avoid any future misunderstanding regarding the evaluation setting that we used and to make sure that all details are clear (and present upfront in the main text itself). We hope our clarifications answer your concern about the validity of the testing scenario. This discussion has helped us become aware that we had not made our testing scenario clear enough in our main text, and has helped us improve our exposition, thank you.

---

> > > > > > > > ### Comment · Reviewer_MQtc · 2023-08-11
> > > > > > > >
> > > > > > > > I see. Please excuse my missing the information in the main rebuttal. This indeed clarifies all my doubts. I will update the score accordingly .

---

### Official Review · Reviewer_kRpA · 2023-07-05

**Soundness:** 3 good
**Presentation:** 3 good
**Contribution:** 3 good
**Rating:** 7
**Confidence:** 4

**Summary:**

This paper proposes a method called Datafix that tries to identify and correct feature shifts in datasets.

This method is composed of two distinct algorithms
DataFix-locate which first identifies what features have shifted between datasets and Datafix-Correct which tries to correct for these features shifts.


Datafix-locate learns a random forest discriminator which is trained to classify reference data from query data. Here the reference data is the non-shifted data while the query data is the data to be tested and corrected for feature shifts.
 This discriminator classifier is used to obtain an approximation to the likelihood ratio between the reference and query data distributions. This approximation is then used to estimate Total Variation (TV) distance between the reference and query data.
The mean decrease of impurity for the random forest classifier is used to estimate importance scores for all features within the data. The top $K$ important features (the top $K$ features whose cumulative importance scores are greater than the scaled TV-distance between the reference and the query datasets) are removed, and the entire procedure is iterated for a set number of times.
All the features throughout this iterative process are the "located features" which have shifted between the reference and the query datasets.

Datfix correct then tries to fix these corrupted or discarded features by first imputing these features through different methods (linear classifiers ,knn,random replacement from reference data) and computes the TV distance between the query data after imputation and t reference data. If this divergence is less than a set threshold, the imputed values are selected as the final corrected values. Otherwise, different "proposals" values for these corrupted features are constructed (which consist of different feature values from reference data as well as previously imputed queries. The proposal value which leads  to the corresponding query having the highest probability of  being a non corrupted sample by the discriminant classifier. The selected proposal values are then chosen to replace the corrupted features. This process is repeated until the TV distance between the reference data and the corrected data is less than a threshold.

Extensive experiments are performed to validate this method. The proposed method is able to outperform baselines on both locating the  corrected  manipulated features (measured through F1 score) and correcting these features (measured through divergence measures )

**Strengths:**

- This paper addresses an important problem which is of great significance to the community.
Featureshift detection is an important problem and this paper proposes a promising method to both identify these shifts and correct for these shifts.
- The method is validated on numerous datasets under numerous feature shift settings (though most of these settings are enforced)
- The experimental setting and results also demonstrate that DataFix can even correct for correlated shifts in features
- The proposed method performs much better than existing baselines on correctly identifying and correcting feature shifts

**Weaknesses:**

- I think the presentation of the work can be improved. Particularly the layout of the paper. There is a lot of discussion in the main paper on material that is not essential to the paper. This particularly includes  section 4 (page 4 line 161 ), such as the discussion on f-divergences (yes, granted there are relevant but not sure how much relevant given only TV is used throughout the paper), and the discussion on mutual information (page 5, line 189 onwards), equation (4).
This causes a lot of important material to be pushed to the appendix, particularly how the proposals are generated when correcting for corrupted features, or how  the algorithms for Feature selection and feature correction work (as provided in the details on the appendix).
This, in my opinion, hurts the readability of the paper  and makes the proposed methods appear much more complicated than it is. It's very difficult to understand the paper without going through the details in the appendix. I would encourage authors to consider improving the readability of the paper by moving parts of appendix D and E to the main paper and maybe moving some of the discussion on f-divergences and the relationship between feature selection  and shift localization to the appendix.
 I would be happy to revise my score if there important reasons for or against this suggestion and I look forward to your response.

- Some of the details, such as how the proposal values are constructed, can be a bit confusing In their current form and would request the authors to provide a small example in the paper (or the appendix) to help explain this.

- Metric for evaluating feature correctness. The authors propose using divergence measures to evaluate how close the corrected query values are to the reference values. This might be fair but from a practical standpoint, I think the purpose of correcting for feature shifts is to ensure the performance of a classifier (or some other ML model) is the same on the reference dataset as that on the corrected dataset. I think such classifier results which evaluate the difference between performance before and after correction would have been helpful.
The corrected features might have  small overall divergence between, but I am not sure this guarantees that class conditioned divergence between the reference and the query data is small. I will be happy to raise my score if the authors could elaborate on this choice of evaluation metric for feature correctness.

**Questions:**

- The experiments show that that under different fraction of shifted features (0 to 0.20), Data-Fix performs much better than other methods. How would the performance scale even under more sever shifted features ? When do you think Data-Fix could break down?
-  The paper mentions that GANs are not used to produce the corrected features as they are not suitable for tablular data. But some of the datasets used such as MNIST (image data) and simulated cosine and ploynomial (functional data) are data modalities where NNs can do well. Do you think a GAN based adversarial approach for feature correction could perhaps perform better in these scenarios?
- A lot feature shift scenarios were implemented within the paper. Though, are there feature shift scenarios which are more realistic and are perhaps encountered more often in real world settings? Are there existing datasets where these feature shifts naturally occur (rather than generating feature shifts as done in the paper)?
- The appendix provides details on computational running times for Data-fix. Are there details for the computing infrastructure used for running these experiments in the appendix? I think these details would be helpful to better understand these computational runtimes.

**Limitations:**

The authors mention in the main paper that limitations are discussed in the appendix, but I wasn't able to find discussion on this.
I think any discussion on the limits of DataFix and when it could fail could be helpful for readers and potential users of DataFix.

---

> ### Author Rebuttal · Authors · 2023-08-08
>
> **Dear Reviewer,**
>
> **Thank you for the highly constructive suggestions. We want to address each of the questions and concerns:**
>
>
> *“the presentation of the work can be improved…  lot of discussion in the main paper is not essential … moving parts of appendix D and E to the main paper and maybe moving some of the discussion on f-divergences and the relationship to the appendix. I would be happy to revise my score...”*
>
> **We completely agree that many important details of the method are in the appendix and that non-essential theoretical information could be moved from the main text. Therefore, we are moving material from sections 4 and 5 to the appendix and adding a more detailed description of the methods (currently in the Appendix) into the main text. Furthermore, with the extra available page upon acceptance, we will try to place the algorithm boxes into the main text.**
>
>
> *“how the proposal values are constructed, can be a bit confusing ... would request the authors to provide a small example in the paper”*
>
> **We agree that it can be confusing to follow the details of the methods, therefore, we are including multiple figures (see attached pdf) in the appendix to provide a better intuition of the inner workings of localization (attached fig 1) and correction methods (attached fig 2). We are including two additional diagrams describing the methods, which do not fit in the attached rebuttal pdf.**
>
>
> *“The authors propose using divergence measures to evaluate …, I think the purpose of correcting for feature shifts is to ensure the performance of a classifier is the same on the reference as on the corrected dataset … small overall divergence between, not guarantees that class conditioned divergence is small. I will be happy to raise my score if the authors could elaborate on this”*
>
> **While in localization/detection tasks there are well-stablished evaluation metrics, in the distribution shift removal problem, the evaluation procedures are not so standardized. We selected the given evaluation metrics because (1) we wanted application-independent metrics, as the corrected datasets might be used in many scenarios (statistical analysis, visualization, machine learning). For example, the proposed algorithm is already being used to homogenize biomedical data in which statistics and correlations are estimated for scientific discovery. Furthermore, some of the datasets do not even have semantic labels, making an evaluation with downstream classification/regression tasks impossible.  (2) As both data imputation and shift correction are closely related tasks (each tries to replace missing/corrupted values with new ones), we wanted to adopt metrics from the imputation literature. Recent papers [a] have used W2 as an imputation evaluation metric, and we included estimates of the symmetric KL divergence and the HP-divergence as they are computationally tractable, and appeared to be sensible estimates of the quality of the shift correction results. Note that we don’t include MSE as it doesn’t properly characterize divergence between distributions (see Appendix C). Finally, note that in imputation literature, application-independent metrics are used.**
>
> *“under different fraction of shifted features Data-Fix performs much better. How would the performance scale? When could break down?”*
>
> **It depends a lot on the type of manipulation. For example, if the mean of just one feature is shifted largely, DataFix can detect it without problem, despite only one feature being present. For more challenging manipulations, such as removing the correlation between features, a small number of features can make DataFix fail. The true underlying divergence between distributions (which is typically unknown) will partly indicate when DataFix will fail. Figure 18,19 in the appendix, which includes an example of how the true and predicted divergences evolve through the filtering process, might help to get a better intuition.**
>
> *“GANs are not used.  But some of the datasets used are data modalities where NNs can do well. Do you think a GAN based adversarial approach for feature correction could perform better?”*
>
> **We do include a GAN-based method for feature correction: GAIN, which combines reconstruction and adversarial losses. However, the method did not provide competitive results and did not scale well for more large dimensional datasets.**
>
> *“A lot feature shift scenarios were implemented. are there feature shift scenarios which are  encountered more often? where these feature shifts naturally occur?”*
>
> **We are encountering many of the proposed shifts when using the system in a biomedical application. Features collapsing to 0 or being negated are not uncommon, especially within genomic sequences. Furthermore, bad standardization (e.g., incorrectly encoding features with the metric or imperial system) can lead to mean shifts. We are open to suggestions for other shifts so we can incorporate them in future works.**
>
> *“Are there details for the computing infrastructure in the appendix?”*
>
> **We forgot to add the hardware details in the appendix, thanks for noticing that! All experiments were done with the same compute resources which included an Intel Xeon Gold with 12 CPU cores. We are adding all this information in the appendix.**
>
> *“The authors mention in the main paper that limitations are discussed in the appendix, but I wasn't able to find this”*
>
> **The description of the results includes some discussion of the limitations (e.g., which manipulation types tend to be detected incorrectly). However, we agree that a section in the appendix dedicated to discussing the limitations of the method would benefit the paper. We are including such a section.**
>
> **We hope that by addressing your concerns you would consider raising the score. Thank you for helping us improve the quality of the manuscript!**
>
> **[a] ICML, 2020, Jarrett, Daniel, et al. Hyperimpute: Generalized iterative imputation with automatic model selection.**

---

> > ### Comment · Reviewer_kRpA · 2023-08-12
> >
> > Thank you for your detailed reply. I have increased my score. I would be willing to further increase my score if the authors could further elaborate on my 3rd response.
> >
> > Please see my response below
> >
> > 1. Thank you for considering my suggestions on improving the readability of the paper. I think these changes could greatly help readers better understand this work (and it seems that some of these changes were also suggested by  Reviewer 5LeU )
> > 2. Also, these new figures in the attached pdf file, particularly MNIST example on proposed figure, are helpful in getting a better understanding of the paper.
> > 3. Thank you for the clarification on why divergence measures to evaluate the proposed method's performance, particularly how a few datasets might not make it possible to evaluate classification/accuracy scores.
> > I think a few sentences on this relation between the used evaluation metrics and imputation evaluation metrics in the paper could be helpful.
> > Though I think there are certainly a few datasets where labels are available (MNIST, UCI datasets). Also it is certainly possible to generate labels for synthetic data.  Additionally, there are other imputation methods, particularly for biomedical applications, that report classification/regression results. [A] is just one example of this (and you cite this in related work section on feature shift correction).
> > I also think that the relationship between imputation and feature shift correction is a bit more subtle. Imputation can certainly be considered 'correction', but feature shifts can potentially lead to scenarios that are more involved than missing samples and imputing them. In the introduction (page 2, line 12,13), the relationship between correction and imputation or alignment is made. Alignment is a much more broad concept than the citations [12,13] suggest (which also just use the divergence scores, also both of these papers  are by the same author. Thus the citation for alignment covers a very narrow alignment area).
> > I think feature correction is inherently a necessity for downstream tasks. I mean that is why they are called 'features' in my opinion, features for some downstream task.
> > I realize that the rebuttal period is very brief to conduct experiments on a large scale.
> > Though I think future works on feature correction should report classification/downstream tasks.
> > This is something that community should consider. I would like to know your opinions on this.
> >
> > 4. Thank you for pointing to GAN baseline.
> >
> > [A]Yoon, Jinsung, James Jordon, and Mihaela Schaar. "Gain: Missing data imputation using generative adversarial nets." International conference on machine learning. PMLR, 2018).

---

> > > ### Author Response · Authors · 2023-08-15
> > >
> > > **Thank you for your response and for upgrading the score! Your reviews have been very useful in improving the manuscript!**
> > >
> > > After considering your arguments about downstream evaluation tasks and carefully looking at the GAIN paper, we agree that such evaluation would benefit the quality of the manuscript. Therefore, **we are including an evaluation of downstream classification and regression tasks as a new section in the appendix**.
> > >
> > > Due to time and compute limitations we have only been able to perform the evaluation for two datasets (Musk2 and Energy), but we will expand it to the other datasets that include classification or regression labels. Instead of using linear models as done in the GAIN paper, we use LightGBM Regressor/Classifier for the downstream evaluation because (1) it reflects a more realistic setting in data analysis, and (2) it makes use of higher-order relationships between features, which linear models fail to capture.
> > >
> > > **We train the downstream models with the query datasets and evaluate them in the reference dataset**. We perform the experiment using each query dataset version (the original pre-corrupted dataset, and the correct datasets with each correction method) and evaluate the classifier/regressor when all features are used (100% in the tables below), when only 50% of the non-corrupted features are included, and when 0% of non-corrupted features are included (only the corrected features are used). We report balanced accuracy for classification and root mean square error for regression in the tables below.
> > >
> > >
> > > **Table 1: Downstream classification task with Musk2 dataset, including all corrected features with all non-corrupted features (100%), with half of the non-corrupted features (50%), and without non-corrupted features (0%).**
> > >
> > > | method         | Balanced Accuracy (0%) | Balanced Accuracy (50%) | Balanced Accuracy (100%) |
> > > |----------------|------------------------|-------------------------|--------------------------|
> > > | Original Query | 0.951                  | 0.960                   | 0.964                    |
> > > | DataFix        | 0.902                  | 0.953                   | 0.957                    |
> > > | HyperImpute    | 0.883                  | 0.939                   | 0.958                    |
> > > | LR             | 0.871                  | 0.932                   | 0.938                    |
> > > | ICE            | 0.848                  | 0.928                   | 0.938                    |
> > > | INB            | 0.833                  | 0.946                   | 0.952                    |
> > > | KNN            | 0.815                  | 0.925                   | 0.934                    |
> > > | Sinkhorn       | 0.806                  | 0.948                   | 0.963                    |
> > > | MIRACLE        | 0.786                  | 0.931                   | 0.953                    |
> > > | MLP            | 0.786                  | 0.905                   | 0.934                    |
> > > | MissForest     | 0.779                  | 0.908                   | 0.944                    |
> > > | GAIN           | 0.717                  | 0.887                   | 0.911                    |
> > > | DD             | 0.550                  | 0.930                   | 0.945                    |
> > > | SoftImpute     | 0.508                  | 0.855                   | 0.855                    |
> > > | Mean           | 0.500                  | 0.952                   | 0.961                    |
> > >
> > > &nbsp;
> > >
> > > **Table 2: Downstream regression with Energy dataset.**
> > >
> > > | method         | RMSE (0%) | RMSE (50%) | RMSE (100%) |
> > > |----------------|----------|------------|-------------|
> > > | DataFix        | 0.089    | 0.082      | 0.080       |
> > > | Original Query | 0.093    | 0.084      | 0.082       |
> > > | KNN            | 0.105    | 0.094      | 0.091       |
> > > | Sinkhorn       | 0.109    | 0.091      | 0.085       |
> > > | SoftImpute     | 0.114    | 0.145      | 0.167       |
> > > | INB            | 0.116    | 0.106      | 0.101       |
> > > | Mean           | 0.122    | 0.088      | 0.083       |
> > > | DD             | 0.128    | 0.122      | 0.119       |
> > > | HyperImpute    | 0.130    | 0.122      | 0.123       |
> > > | GAIN           | 0.130    | 0.170      | 0.206       |
> > > | MissForest     | 0.147    | 0.155      | 0.156       |
> > > | LR             | 0.191    | 0.189      | 0.205       |
> > > | MLP            | 0.212    | 0.225      | 0.233       |
> > > | ICE            | 0.217    | 0.225      | 0.234       |
> > > | MIRACLE        | 0.334    | 0.350      | 0.352       |
> > >
> > >
> > > Methods are sorted by their performance without non-corrupted features (0%).  The classification/regression downstream performance when using **DataFix surpasses the competing methods** in most settings, in some cases even surpassing the performance of directly using the original query dataset (ground truth pre-corruption).
> > > **We hope that these additional results help to improve the quality of the manuscript and that you will consider further increasing the score. Thank you for all your excellent feedback!**

---

> > > > ### Comment · Reviewer_kRpA · 2023-08-15
> > > >
> > > > Thank you for constructively considering my comments on downstream evaluation metrics.
> > > > Also, thank you for running these classification/regression experiments in a relatively short amount of time. The results on how data fix improves on original query in some settings is certainly interesting.
> > > > I have increased my score.

---

### Official Review · Reviewer_dDZx · 2023-07-07

**Soundness:** 3 good
**Presentation:** 3 good
**Contribution:** 3 good
**Rating:** 7
**Confidence:** 4

**Summary:**

This paper studies the problem when a subset of coordinates in the features have shifts in distributions. It specifically proposes algorithms to detect the shifts, and methods to “repair” the shifted coordinates.

Imo, the most interesting part of the paper is that it relates certain inequalities in divergence measures with GAN via likelihood ratio test. Under this interpretation, the problem of identifying distributional shifts becomes tuning some classifiers. Their experiments also confirm that this approach is effective.

Then the second part of repairing the corrupted coordinates smell a bit hackish, e.g., it proposes a few candidate solutions, then it runs some classification algorithms to determine which candidates to use. Nevertheless, it appears to me that the central idea is to use unchanged features to “complete” the changed ones, which I believe is plausible. In general, this part feels heavy lifting and I do not have much intuition on that.

Overall, I believe this is a quite solid contribution to neurips.



**Strengths:**

it makes an interesting connection between distributional shift and GAN.

**Weaknesses:**

Second part sounds a bit hackish. I also have a few questions (see below).

**Questions:**


I have a few major concerns/questions.

This work seems not very careful on the expository text. For example, statistical distinguishable does not imply computational distinguishable (foundation of modern/any cryptography?). So D(p, q) > \epsilon does not always mean that they can be detected.
Divergence measures are asymmetric. Can you comment about this, e.g., do we have two thresholds or always put the original distribution to the left?
When we can “complete” the features with shifted distributions for training and/or inference purpose, does that mean these features are not important and can be excluded anyway?


**Limitations:**

NA.

---

> ### Author Rebuttal · Authors · 2023-08-08
>
> **Dear Reviewer,**
>
> **Thank you for the thoughtful review! We agree that some of the introduced heuristics can feel a “bit hackish”, however, they have proved to be more accurate than previous works and than our attempts with more conventional supervised learning methods. One way to get a better intuition for the correction system is to compare it with a k-NN imputer: a k-NN imputer fills the missing feature values with a combination of the top-k closest samples with respect to L2 distance on non-missing features.** **Our correction system replaces the corrupted feature values with the top-1** **“candidate” with respect to the probability estimated by the discriminator.** **In a sense, the correction task is a divergence-reducing imputation problem.** **We are including a more in-depth discussion, as well as additional figures (see attached pdf), to provide better intuition to the reader and to further motivate our approach.**
>
> &nbsp;
>
> *“statistical distinguishable does not imply computational distinguishable”*
>
> **Definitely, thank you for catching this error, we have re-phrased the text to avoid making this mistake.**
>
> &nbsp;
>
> *“Divergence measures are asymmetric.”*
>
>
> **While many f-Divergences are asymmetric (e.g. KL-Divergence), there are a few that are in fact symmetric, such as the “Hellinger distance” or the “Total Variation” used in this work.**
>
>
> &nbsp;
>
> *“When we can “complete” the features with shifted distributions for training and/or inference purpose, does that mean these features are not important and can be excluded anyway?”*
>
> **Deciding to use corrected features or simply discarding them will depend largely on the application at hand. For example, this method (using corrections) is currently being deployed for biomedical applications (for instance genomics), where simple statistics and correlations are desired from the features for scientific discovery. Using corrected features allows one to use all of the data, while not distorting the statistical estimates as severely as using the corrupted data would.**

---

### Author Rebuttal · Authors · 2023-08-09

**We would like to thank all reviewers for their constructive comments,**

**First, we are restructuring the text to improve the readability of the paper. As suggested by the reviewers, we are including more details about the proposed methods in the main text, and moving some theoretical results to the appendix. Additional figures (fig 1,2,5 in the attached document) are included in the appendix providing better intuition for the design details of the methods. Furthermore, we are including in both the main text and appendix more details regarding the evaluation setting, hyperparameter search, and computational resources, as well as an additional figure with error bars showing the variability between random seeds (see fig 3,4 in the attached pdf).**

**We want to clarify the motivation for the proposed work: while presented jointly, both the localization and correction work can be useful by themselves. The shift localization can be used to detect incorrectly formatted features, errors arising during data processing, or in multi-dimensional sensor applications, detecting malfunctioning sensors. The work presented in [Neurips 2020] provides a similar setting and set of motivations and acts as our benchmark baseline. The shift correction system can be used in any application where simple imputation would be used, providing a replacement for corrupted/missing values that minimize an empirical divergence. Deciding to use one or both systems is completely dependent on the application. We are modifying the text accordingly to properly depict the motivation of the work.**

**While the proposed methods can be used to extend training datasets for training other ML models (e.g., supervised classification or regression), they are not limited to such tasks, and in fact, many of the datasets used do not have labels, making an evaluation using downstream ML classifiers impossible. In fact, our system is currently being deployed in a biomedical setting in order to homogenize data that is later used to compute statistics and correlations between features for scientific discovery. Therefore, we adopt a more appropriate evaluation setting, independent of the application where DataFix is applied, as is commonly done in related tasks such as data imputation [ICML 2020]. Finally, we want to note that the evaluation setting adopted is correct, no train/test split is required (as in data imputation tasks), and we have a set of datasets (simulated data) for hyperparameter tuning and a set for purely testing (real datasets). We are modifying the text accordingly to properly justify the adopted evaluation setting.**

**The attached pdf with figures include:**
- Figure 1: examples of feature importance during the iterative localization process with different parameters.
- Figure 2: number of features filtered at each iteration, with different hyperparameters.
- Figure 3: error bars for shift correction.
- Figure 4: error bars for shift localization.
- Figure 5: example of proposals in MNIST dataset for shift correction.

**Two extra diagrams providing more details of our proposed methods are included in the appendix.**

**We hope that we have addressed all of your concerns. As the method provides accurate results, it is useful in many applications (in fact, it is already being deployed), and your suggestions have helped us to largely improve the text and structure of the manuscript, we believe that our work is a strong submission to NeurIPS, and we hope that you will consider raising the score.**

**[NeurIPS 2020] Kulinski, Sean, Saurabh Bagchi, and David I. Inouye. "Feature shift detection: Localizing which features have shifted via conditional distribution tests."**

**[ICML 2020], Jarrett, Daniel, et al. Hyperimpute: Generalized iterative imputation with automatic model selection.**

---

### Decision · Program_Chairs · 2023-09-21

**Decision:**

Accept (poster)

**Comment:**

My recommendation is to accept the paper.

After active discussion with the authors, reviewers agreed that the paper makes a valuable contribution. It does seem like some edits are in order for the camera ready to clarify points that reviewers found confusing during the review and discussion periods, but these seem very feasible post-review. Please look over the reviewer suggestions carefully and incorporate them (some of this is already promised in the rebuttal).